# Structural analysis of autoinhibition in the Ras-specific exchange factor RasGRP1

Jeffrey S Iwig[1,2], Yvonne Vercoulen[3†], Rahul Das[1,2†], Tiago Barros[1,2,4], Andre Limnander[3], Yan Che[1,2], Jeffrey G Pelton[2], David E Wemmer[2,5,6], Jeroen P Roose[3]*, John Kuriyan[1,2,4,5,6]*

[1]Department of Molecular and Cell Biology, University of California, Berkeley, Berkeley, United States; [2]California Institute for Quantitative Biosciences, University of California, Berkeley, Berkeley, United States; [3]Department of Anatomy, University of California, San Francisco, San Francisco, United States; [4]Howard Hughes Medical Institute, University of California, Berkeley, Berkeley, United States; [5]Department of Chemistry, University of California, Berkeley, Berkeley, United States; [6]Physical Biosciences Division, Lawrence Berkeley National Laboratory, Berkeley, United States

**Abstract** RasGRP1 and SOS are Ras-specific nucleotide exchange factors that have distinct roles in lymphocyte development. RasGRP1 is important in some cancers and autoimmune diseases but, in contrast to SOS, its regulatory mechanisms are poorly understood. Activating signals lead to the membrane recruitment of RasGRP1 and Ras engagement, but it is unclear how interactions between RasGRP1 and Ras are suppressed in the absence of such signals. We present a crystal structure of a fragment of RasGRP1 in which the Ras-binding site is blocked by an interdomain linker and the membrane-interaction surface of RasGRP1 is hidden within a dimerization interface that may be stabilized by the C-terminal oligomerization domain. NMR data demonstrate that calcium binding to the regulatory module generates substantial conformational changes that are incompatible with the inactive assembly. These features allow RasGRP1 to be maintained in an inactive state that is poised for activation by calcium and membrane-localization signals.

*For correspondence: jeroen. roose@ucsf.edu (JPR); kuriyan@ berkeley.edu (JK)

†These authors contributed equally to this work

## Introduction

An intriguing aspect of lymphocyte development is that the generation of a population of self-tolerant immune cells requires Ras activation by two distinct guanine nucleotide exchange factors, Ras guanine nucleotide releasing protein 1 (RasGRP1) and Son-of-Sevenless (SOS) (*Figure 1A*) (*Dower et al., 2000*; *Ebinu et al., 2000*; *Kortum et al., 2011*). Ras cycles between an inactive GDP-bound form and an active GTP-bound form, and in both states the nucleotide is very tightly bound (*Vetter and Wittinghofer, 2001*; *Rajalingam et al., 2007*; *Ahearn et al., 2012*). A critical regulatory function is therefore provided by the action of guanine nucleotide exchange factors, which loosen the grip of Ras on the bound nucleotide, allowing GTP loading and activation (*Bos et al., 2007*; *Cherfils and Zeghouf, 2013*).

There are three major families of Ras-specific nucleotide exchange factors in humans. The RasGRP and Ras guanine nucleotide releasing factor (RasGRF) families of exchange factors have tissue-specific expression patterns whereas SOS proteins are expressed ubiquitously (*Stone, 2011*). RasGRP proteins have been studied most extensively in T and B lymphocytes (*Aiba et al., 2004*; *Brodie et al., 2004*; *Coughlin et al., 2005*; *Roose et al., 2005*; *Limnander et al., 2011*) where they activate Ras in a manner that is non-redundant with SOS (*Dower et al., 2000*; *Roose et al., 2007*). In addition, RasGRP proteins play important roles in squamous cell carcinoma and melanoma (*Luke et al., 2007*; *Oki-Idouchi and Lorenzo, 2007*; *Diez et al., 2009*; *Yang et al., 2011*), T cell- and myeloid- leukemia

**eLife digest** Individual cells within the human body must grow, divide or specialize to perform the tasks required of them. The fates of these cells are often directed by proteins in the Ras family, which detect signals from elsewhere in the body and orchestrate responses within each cell. The activities of these proteins must be tightly controlled, because cancers and developmental diseases can result if Ras proteins are not properly regulated.

Binding to the small molecule GTP activates Ras and causes conformational changes that allow it to interact with other proteins in various signaling pathways in the cell. GTP is loaded into Ras by proteins called nucleotide exchange factors, which can replace 'used' nucleotides with 'fresh' ones to activate Ras.

These nucleotide exchange factors are also tightly regulated. For example, the genes for many exchange factors are only switched on after particular signals are received, which can restrict their presence to defined times and locations (e.g., cells or tissues). Also, when activating signals are absent, nucleotide exchange factors commonly reside in the cytoplasm, whereas the Ras proteins remain bound to lipid membranes inside the cell.

RasGRP1 is a nucleotide exchange factor that controls the development of immune cells, and leukemia and lupus can result if it is not regulated correctly. However, many questions about RasGRP1 remain unanswered, including how it is able to remain inactive, and how it is activated by various different signals.

Iwig et al. have now revealed the mechanisms through which RasGRP1 suppresses Ras signaling in immune cells by solving the structures of two fragments of RasGRP1 and then using a combination of structural, biochemical and cell-based methods to explore how it is activated. These analyses revealed that inactive RasGRP1 adopts a conformation in which one of its regulatory elements blocks access to the Ras binding site. Surprisingly, RasGRP1 can form dimers; this hides the portions of the protein that associate with the membrane and thereby keeps RasGRP1 away from Ras. Iwig et al. also found that two signals, calcium ions and a lipid called diacylglycerol, overcome these inhibitory mechanisms by changing the conformation of RasGRP1 and recruiting it to the membrane.

These studies provide a framework for understanding how disease-associated mutations in RasGRP1 bypass the regulatory mechanisms that insure proper immune cell development, and will be critical for developing therapeutic agents that inhibit RasGRP1 activity.

(*Reuther et al., 2002*; *Yang et al., 2002*; *Klinger et al., 2005*; *Lauchle et al., 2009*; *Oki et al., 2012*; *Hartzell et al., 2013*) and prostate cancer (*Yang et al., 2010*) that are distinct from those of SOS.

Developing T lymphocytes pass through quality control checkpoints to generate a repertoire of protective but self-tolerant immune cells (*Starr et al., 2003*) and Ras signaling plays a critical role in this progression (*Swan et al., 1995*). In response to T cell receptor stimulation during T lymphocyte development, both RasGRP1 and SOS are recruited to the membrane where they encounter membrane-anchored Ras. Interestingly, knockout mouse models have revealed that the requirements for RasGRP1 and SOS during different T lymphocyte checkpoints are distinct, despite the fact that they both convert Ras•GDP to Ras•GTP (*Dower et al., 2000*; *Layer et al., 2003*; *Kortum et al., 2011*, *2012*).

The importance of RasGRP1 in human diseases highlights a critical requirement for tight regulation of RasGRP1 activity. Elevated *RasGRP1* mRNA expression has been reported in T cell leukemia microarray studies and is found frequently in pediatric T cell leukemia in which it stimulates the growth of this blood cancer (*Oki et al., 2012*; *Hartzell et al., 2013*). Conversely, reduced RasGRP1 expression has been reported for autoimmune patients with lupus erythematosus where it may play a role in aberrant DNA methylation in T cells (*Yasuda et al., 2007*; *Pan et al., 2010*). Additionally, single nucleotide polymorphisms in *RasGRP1* have been described in genome-wide association studies of autoimmune diabetes and thyroid disease (*Qu et al., 2009*; *Plagnol et al., 2011*).

The Ras-specific exchange factors have similar catalytic modules that contain two domains. The Cdc25 domain interacts directly with Ras and dislodges the bound nucleotide (*Boriack-Sjodin et al., 1998*). The Ras exchanger motif (REM) domain that is associated with the Cdc25 domain is usually essential for activity but its function does not appear to be conserved in different exchange factors.

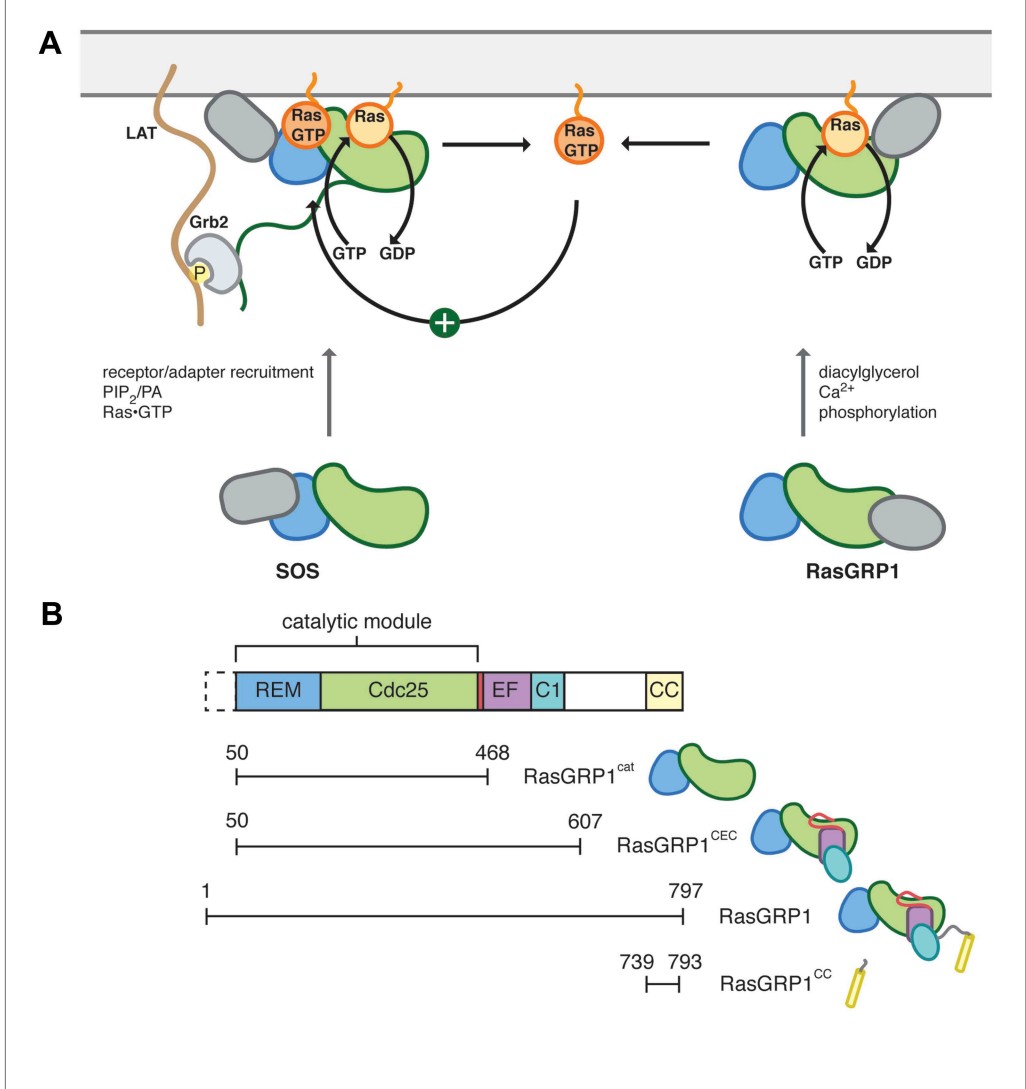

**Figure 1**. Control of Ras activity in T cells. (**A**) Ras cycles between an inactive, GDP-bound form and an active GTP-bound form. In T cells, two nucleotide exchange factors, SOS and RasGRP1 enhance the removal of nucleotide from Ras, which is then replaced with GTP. Each exchange factor shares a common catalytic module but is regulated by distinct signaling inputs. SOS activity is enhanced by Ras•GTP, generated by RasGRP1, binding to an allosteric site. The regulatory domains from each exchange factor are distinct and are represented in gray. SOS is recruited to the membrane in part by Grb2, which interacts with phosphotyrosine residues in the adapter LAT. (**B**) The catalytic core of RasGRP1 includes the REM and Cdc25 domains, which are followed by a regulatory module containing the EF domain, membrane binding C1 domain and a predicted coiled coil. An alternate translational start site is present that leads to a RasGRP1 protein without the first 49 residues. The constructs used in this study are shown.

The following figure supplements are available for figure 1:

**Figure supplement 1**. Domain architecture of RasGRP1 and SOS.

---

Each family of Ras-specific exchange factors contains distinct regulatory domains that enable Ras signaling to be activated in response to a variety of upstream receptor stimuli. Despite the importance of the regulatory domains for controlling activation, our understanding of how these work at the structural level is limited to SOS (*Sondermann et al., 2004*; *Gureasko et al., 2008*, *2010*) and the Rap-specific exchange factor, Epac2 (*Rehmann et al., 2006*, *2008*).

One important role for RasGRP1 is to prime SOS for activation by generating an initial burst of Ras•GTP (*Roose et al., 2007*). This priming function of RasGRP1 potentiates SOS activity because of

a feedback loop in which Ras•GTP activates SOS by binding to an allosteric site that bridges the REM and Cdc25 domains (*Margarit et al., 2003*; *Boykevisch et al., 2006*; *Sondermann et al., 2004*; *Gureasko et al., 2008*, *2010*). Ras•GTP binding to the allosteric site helps stabilize SOS at the plasma membrane and promotes the conversion of Ras•GDP to Ras•GTP. The action of RasGRP1 in initiating the positive feedback loop of SOS leads to ultrasensitive ERK activation in Jurkat T cells and has been postulated to define the sharp boundary between positively and negatively selecting ligands during thymocyte development (*Das et al., 2009*; *Prasad et al., 2009*). Compartmentalization of Ras signaling has also been proposed to play a role in the selection process (*Daniels et al., 2006*). A complete understanding of how the interplay between RasGRP1 and SOS results in ultrasensitive activation of the ERK pathway requires mechanistic knowledge of how RasGRP1 is regulated, about which little is known.

The catalytic module of RasGRP1 is followed by an EF domain with a predicted pair of EF hands (EF1 and EF2 modules), a diacylglycerol-binding C1 domain, and a C-terminal segment that includes a primarily unstructured region of ~140 residues and a predicted coiled coil (*Ebinu et al., 1998*; *Beaulieu et al., 2007*; *Zahedi et al., 2011*) (see *Figure 1B* for the domain architecture of RasGRP1). A portion of the C-terminal segment of RasGRP1 has been demonstrated to enhance membrane recruitment through electrostatic interactions with phosphoinositides (*Zahedi et al., 2011*), and the physiological importance of this segment is illustrated by impaired T lymphocyte development in mice lacking this part of the protein (*Fuller et al., 2012*).

Little is known about how the regulatory domains of RasGRP1 control the activity of the catalytic module. The simplest model for RasGRP1 activation assumes that the recruitment of the protein from the cytosol to the membrane upon diacylglycerol production by phospholipase C suffices for activation by facilitating encounters with Ras. However, addition of a membrane localization tag to a fragment of RasGRP1 does not lead to constitutive Ras activation, suggesting more complexity in the regulatory mechanisms (*Beaulieu et al., 2007*). The presence of two EF hands suggests that they might be responsible for the sensitivity of RasGRP1 to calcium, but there are conflicting reports as to whether calcium binding to the EF domain is coupled to the localization and activity of RasGRP1 (*Ebinu et al., 1998*; *Lorenzo et al., 2000*; *Tazmini et al., 2009*).

To identify the structural basis for the regulation of RasGRP1, we have determined two crystal structures of RasGRP1. Together, these structures span the folded domains of the protein and omit the N-terminal 50 residue segment and the ~140 residue segment immediately following the C1 domain that are both predicted to be intrinsically disordered. The first structure includes the REM, Cdc25, EF and C1 domains and suggests a structural basis for autoinhibition by the regulatory domains. Key aspects of the mechanisms we identify involve occlusion of the Ras-binding site by an interdomain linker and dimerization-mediated masking of the C1 domains. The second structure is pertinent in this regard, as it shows that the C-terminal segment forms a parallel coiled coil that facilitates dimerization. The EF domain, which we demonstrate has a single $Ca^{2+}$ binding site, is likely to contribute to activation through a calcium-triggered conformational change that we have analyzed using NMR. Our results are consistent with a mechanism wherein membrane docking and calcium binding drive RasGRP1 activation by disrupting the C1 interface and removing an inhibitory linker that blocks the Ras-binding site.

## Results and discussion

### Structures of RasGRP1

#### Structure of the autoinhibited catalytic module

Efforts to express and purify full-length human RasGRP1 (797 residues) from *Escherichia coli* (*E. coli*) or insect cells were unsuccessful. We found instead that an N-terminal fragment (residues 50–607, RasGRP1[CEC]) and a C-terminal fragment (residues 739–793, RasGRP1[CC]) could be expressed separately in *E. coli* in soluble forms (see *Figure 1B* for nomenclature). These segments are separated by a ~140 residue linker in the intact protein that is likely to be predominantly disordered based on secondary structure predictions. We determined a crystal structure of RasGRP1[CEC] at 3.0 Å resolution (*Table 1*, *Figure 2A*). The catalytic modules of RasGRP1 and SOS are structurally similar. In contrast, the structures of the regulatory domains and their inhibitory interactions with the catalytic module are different in RasGRP1 and SOS.

The Cdc25 domain consists of a compact bundle of 10 helices that forms the core of the structure. Two antiparallel and tightly packed helices that form a prominent hairpin protrude from this core. The

**Table 1.** Structure determination and refinement of RasGRP1$^{CEC}$

| Data collection | | | |
|---|---|---|---|
| | Native | Zn$^{2+}$ | SeMet |
| Wavelength (Å) | 1.000 | 1.283 | 0.979 |
| Space group | P6$_1$22 | P6$_1$22 | P6$_1$22 |
| Cell dimensions | | | |
| $a,b,c$ (Å) | 76.3, 76.3, 408.4 | 77.2, 77.2, 413.4 | 76.3, 76.3, 410.3 |
| Resolution (Å) | 47.4–3.0 | 48.0–3.3 | 47.5–3.5 |
| $R_{sym}$ (%) | 11.1 (94.9) | 9.7 (70.7) | 14.4 (98.0) |
| $I/\sigma(I)$ | 10.9 (1.5) | 12.4 (2.3) | 11.8 (2.4) |
| $CC_{1/2}$ | 99.9 (53.3) | 99.7 (47.7) | 99.9 (57.9) |
| Completeness (%) | 99.7 (99.8) | 99.7 (99.7) | 99.6 (97.9) |
| Redundancy | 7.4 (5.2) | 4.9 (5.1) | 9.2 (9.1) |
| Wilson B factor | 74.50 | | |
| **Refinement** | | | |
| Resolution | 47.4–3.0 | | |
| Reflections used | 15146 | | |
| $R_{free}$ reflections | 774 | | |
| $R_{work}/R_{free}$ | 0.224/0.269 | | |
| No. Atoms | | | |
| Protein | 4223 | | |
| Ligands | 21 | | |
| Average $B$ factors | | | |
| Protein | 108.4 | | |
| Solvent | 54.0 | | |
| Root mean square deviation from ideality | | | |
| Bonds (Å) | 0.007 | | |
| Angles (°) | 1.020 | | |
| Ramachandran statistics | | | |
| Favored (%) | 97.9 | | |
| Disallowed (%) | 0.0 | | |
| MolProbity clash score | 21.3 | | |

structure of nucleotide-free Ras bound to the SOS Cdc25 domain showed that the switch 2 region of Ras docks on the helical bundle of the Cdc25 domain, and that the helical hairpin splays the switch 1 segment of Ras away from the rest of the protein, thereby removing the nucleotide (*Figure 3A*) (*Boriack-Sjodin et al., 1998*; *Margarit et al., 2003*). A similar interaction between Rap1 and the Cdc25 domain was seen in Epac2 (*Rehmann et al., 2008*). Based on the close structural similarity of the Cdc25 domains, empty Ras is expected to bind the Cdc25 domain of RasGRP1 in a manner similar to that observed for Ras binding to SOS (*Figure 3A*). The hydrophobic properties of residues in SOS that form a pocket at the base of the helical hairpin and interact with Tyr 64 and the switch 2 element of Ras (*Boriack-Sjodin et al., 1998*) are conserved in RasGRP1. These residues include Ile 825, Phe 929 and Leu 872 in SOS.

The REM domain of RasGRP1 interacts with the helical hairpin of the Cdc25 domain in a manner similar to that seen in SOS (*Figure 3A*). Despite the limited sequence identity between the REM domains of RasGRP1 and SOS, the folds are similar with the exception of a helical extension in SOS that is absent in RasGRP1. The REM domain extension in SOS helps orient the N-terminal regulatory module (see *Figure 1—figure supplement 1* for SOS domain architecture), which occludes the allosteric Ras-binding site in the autoinhibited state (*Sondermann et al., 2004*; *Gureasko et al., 2010*).

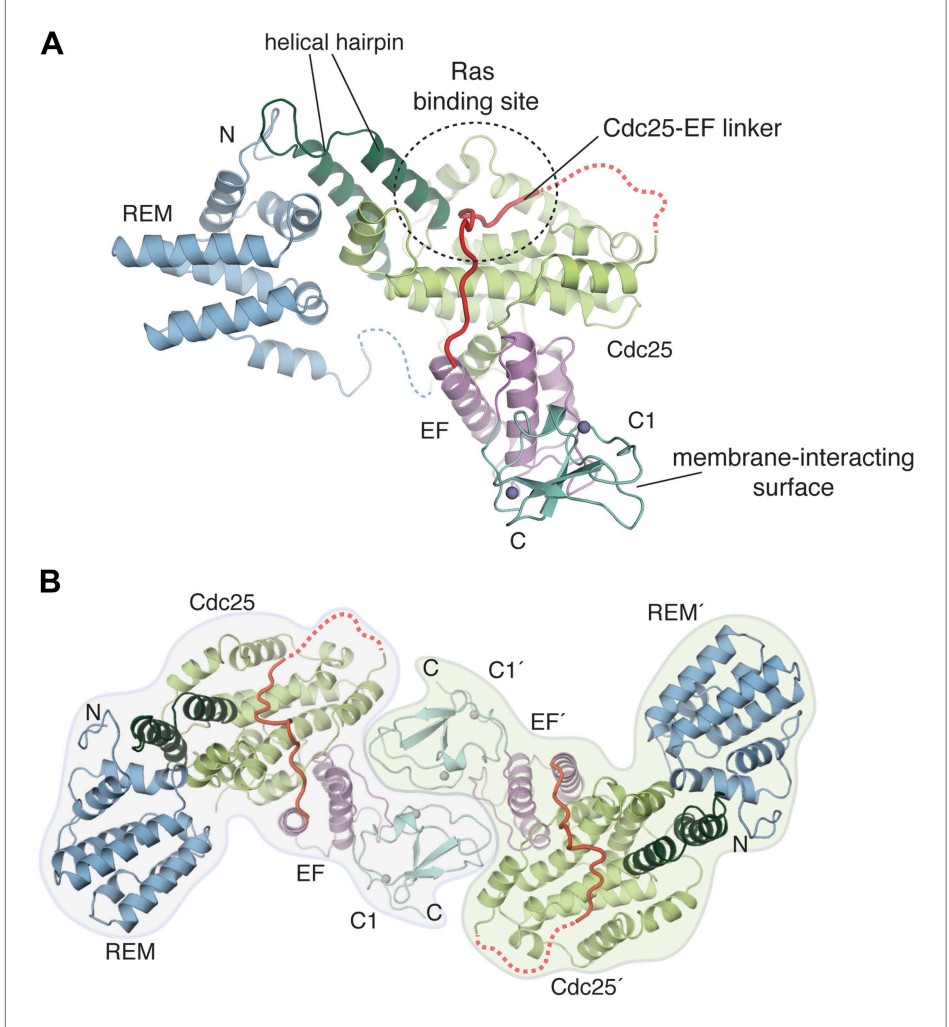

**Figure 2**. Crystal structure of the RasGRP1 autoinhibited catalytic module. (**A**) A crystal structure of the first four domains of RasGRP1 shows the REM domain (blue) buttressing the helical hairpin of the Cdc25 domain (green). The EF domain (magenta) is sandwiched between one side of the Cdc25 domain and the C1 domain (teal). Two zinc ions in the C1 domain are shown as gray spheres. The Cdc25-EF linker (red) traverses the Ras-binding site on the Cdc25 domain. Linkers that could not be modeled due to poor electron density are shown with dotted lines. The N- and C-termini are indicated by N and C, respectively. (**B**) The C1 domain mediates formation of a crystallographic dimer. The domains of one monomer are denoted with primes.

The following figure supplements are available for figure 2:

**Figure supplement 1**. The Cdc25-EF linker occupies the Ras binding site in the Cdc25 domain.

---

The orientation of the helical hairpin in SOS is controlled by the REM domain, which is in turn responsive to the binding of Ras•GTP at the allosteric site.

In the absence of Ras•GTP at the allosteric site, the helical hairpin of SOS adopts a closed conformation that partially occludes the Ras binding site. A ~10° rotation of the SOS helical hairpin is necessary to achieve a conformation competent for Ras binding to the active site. In contrast, the RasGRP1 helical hairpin (residues 355–398) in the absence of Ras is rotated ~15° away from the active site with respect to its orientation in active SOS, which is consistent with an open conformation that is poised to bind to Ras (*Figure 3B*). This critical difference in the helical hairpin orientation implies that RasGRP1 regulation may not rely on allosteric control using the REM domain. The helical hairpin conformation for RasGRP1 is similar to the orientation observed for the isolated Cdc25 domain of RasGRF (*Freedman et al., 2006*). For Epac2, the helical hairpin is in an open conformation in the absence of Rap (*Rehmann et al., 2006, 2008*).

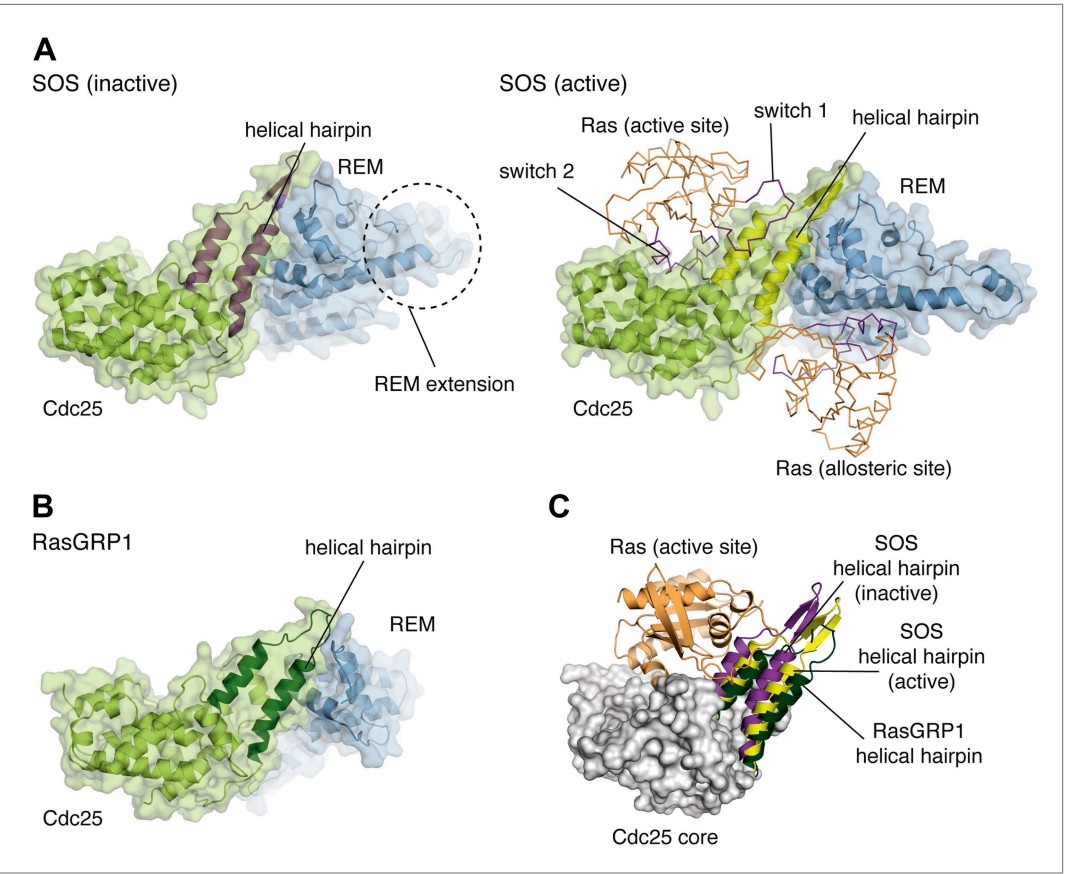

**Figure 3**. Comparison of the catalytic modules of SOS and RasGRP1. (**A**) Structures of SOS<sup>cat</sup> in the inactive state (PDB ID:2II0) (left) and active state (PDB ID: 1NVV) (right) bound to Ras (orange) at the active and allosteric sites are shown. The switch 1 and switch 2 elements of Ras are shown in purple. (**B**) The architecture of the catalytic module of RasGRP1 is similar to that of SOS, indicating that active site Ras will bind to at a similar location in the Cdc25 domain. (**C**) The helical hairpins of inactive SOS (purple), active SOS (yellow) and RasGRP1 (green) are shown with the core of the RasGRP1 Cdc25 domain (gray) and Ras modeled at the active site of RasGRP1 (orange). The helical hairpin of RasGRP1 is rotated ~25° away from Ras relative to the helical hairpin of SOS in the inactive state. In this conformation, the helical hairpin of SOS occludes the Ras binding site.

The regulatory module of RasGRP1, containing the EF and C1 domains, interacts extensively with the Cdc25 domain. This is in contrast to the situation with SOS, where a different set of regulatory domains is organized around the REM domain, making no contact with the Cdc25 domain. The EF domain in RasGRP1 interacts with one side of the Cdc25 domain, adjacent to the catalytic Ras-binding site, with the C1 domain extending from the opposite side of the EF domain. Metal ions are not present in either EF hand in the crystal structure. The second of the two EF hands is structurally degenerate in that one of the two helices present in canonical EF hands is missing and is replaced by a loop that we refer to as the EF2 connector.

The RasGRP1 C1 domain is structurally similar to other C1 domains, with a Cα rmsd of 0.4 Å over 52 aligned Cα atoms when compared with the C1 domain of PKCδ (*Zhang et al., 1995*). The C1 domain docks onto a small region of the EF domain using the surface distal from the membrane-interacting residues. An extensive dimer interface is generated at a twofold symmetry axis in the crystal, burying ~2500 Å² of surface area between the two subunits in the dimer (*Figure 2B*). The C1 domain from one subunit in the dimer interacts with the Cdc25, EF and C1 domains of the other subunit.

### The C-terminal oligomerization domain forms a parallel coiled coil
The RasGRP1 C-terminal region is important for function (*Fuller et al., 2012*) and is unique to RasGRP1 proteins found in higher vertebrates. Residues 747–789 within the C-terminal region are predicted by the COILS webserver (*Lupas et al., 1991*) to contain a coiled coil and are spanned by the RasGRP1<sup>CC</sup>

construct. We determined the crystal structure of RasGRP1[CC] at 1.6 Å resolution. Two molecules in the asymmetric unit of the crystal form a parallel, dimeric coiled coil of nearly 10 helical turns that is structurally similar to canonical coiled coil proteins such as GCN4 (*O'Shea et al., 1991*) (*Figure 4A*, *Table 2*). Leucine and isoleucine residues stabilize the dimer, and a pair of glutamine residues in the core maintains the proper sequence register between the two helices. Two pairs of ionic interactions (Glu 754-Lys 759' and Glu 782-Lys 783', where the primes refer to the second subunit in the dimer) further stabilize the dimer. Together, the dimeric structures of the autoinhibited catalytic module and the C-terminal oligomerization domain provide a framework for understanding the regulatory mechanisms of full length RasGRP1 (*Figure 4B*).

## Mechanisms of RasGRP1 autoinhibition

### The Cdc25-EF linker inhibits nucleotide exchange activity

An electron density map calculated with phases derived from the model for the four domains of RasGRP1[CEC] reveals additional electron density for a peptide traversing the Ras binding site in the Cdc25 domain (*Figure 2—figure supplement 1*). This electron density can be assigned unambiguously to a portion of the linker between the Cdc25 and EF domains, but strong electron density is present only for the central eight residues of the linker. We have identified a plausible sequence register for this region that is shown in *Figure 5A*. Although the electron density for the peptide backbone is strong, there is some uncertainty as to the accuracy of the sequence register within the linker because electron density for sidechains in this segment is not well resolved.

The visible portion of the linker shows that several residues, including Val 450, Val 452 and Trp 454, dock within a hydrophobic groove at the base of the RasGRP1 helical hairpin (*Figure 5A*). This groove is the expected binding site for the switch 2 segment of Ras, including Tyr 64, which is absolutely required for recognition of Ras by SOS (*Hall et al., 2001*). Clearly, occupation of this site on the Cdc25 domain by the Cdc25-EF linker will prevent Ras binding and nucleotide exchange. Trp 454 and the hydrophobic nature of this region of the linker are highly conserved in all RasGRP family proteins, including those in distantly related organisms such as *Caenorhabditis elegans* (*Figure 5B*).

We first tested the effect of mutations in the Cdc25-EF linker on nucleotide exchange activity using an in vitro assay in which the release of labeled nucleotide from Ras is monitored by fluorescence as a function of time (*Ahmadian et al., 2002*). Mutation of Trp 454 and other conserved linker residues leads to a ~2.5-fold increase in RasGRP1[CEC] activity towards soluble Ras in vitro (*Figure 5C*). Mutations in the linker have minimal effect in a construct lacking the EF and C1 domains (data not shown), presumably because the loss of the C-terminal tether of the Cdc25-EF linker results in increased flexibility. Consistent with this idea, RasGRP1[CEC] is ~3.5-fold less active than the isolated catalytic module (REM + Cdc25, RasGRP[cat]) towards Ras in solution (*Figure 5C*).

We next tested the effect of mutations in the Cdc25-EF linker in the context of the full-length protein in intact cells by monitoring the levels of phosphorylated ERK (P-ERK), using fluorescence activated cell sorting (FACS), as a measure of Ras activation (*Figure 5D*). These experiments were performed using Jurkat T lymphoma cells in which the expression of endogenous RasGRP1 is reduced to ~10% of the normal levels (*Roose et al., 2005*),

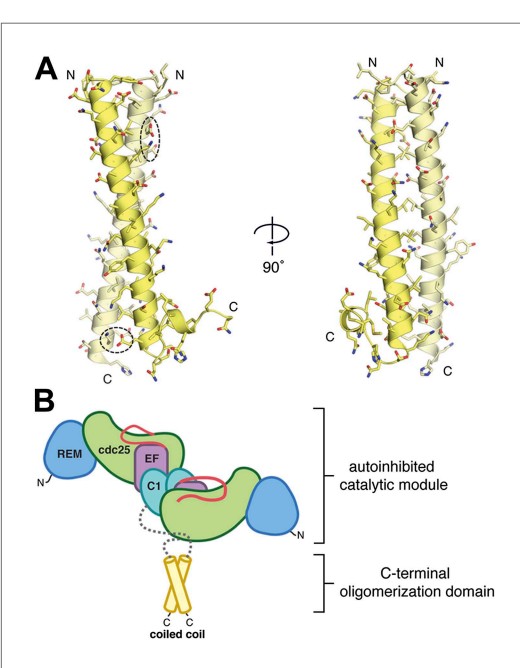

**Figure 4**. The C-terminal domain mediates oligomerization. (**A**) A parallel, dimeric coiled coil is formed by ~40 residues/monomer near the C-terminus of RasGRP1. The individual monomers within the dimer are colored yellow and light yellow. Two pairs of bridging ionic interactions (dotted ellipses), and a Leu/Ile-rich core stabilize the dimer. (**B**) The coiled coil may stabilize the autoinhibited catalytic module. The ~140 residues between the C1 domain and the coiled coil (gray) are predicted to be primarily unstructured.

**Table 2.** Structure determination and refinement of RasGRP1$^{CC}$

| Data collection | |
| --- | --- |
| | Native |
| Wavelength (Å) | 1.000 |
| Space group | P2$_1$2$_1$2 |
| Cell dimensions | |
| a,b,c (Å) | 24.88, 165.05, 28.32 |
| Resolution (Å) | 27.9–1.6 |
| $R_{sym}$ (%) | 5.1 (71.7) |
| $I/\sigma(I)$ | 13.2 (1.8) |
| $CC_{1/2}$ | 99.9 (58.3) |
| Completeness (%) | 98.9 (97.4) |
| Redundancy | 3.0 (3.1) |
| Wilson B factor | 16.51 |
| **Refinement** | |
| Resolution | 27.9–1.6 |
| Reflections used | 16048 |
| $R_{free}$ reflections | 807 |
| $R_{work}/R_{free}$ | 0.201/0.235 |
| No. Atoms | |
| Protein | 797 |
| Ligands | 23 |
| Water | 83 |
| Average B factors | |
| Protein | 32.9 |
| Solvent | 42.8 |
| Root mean square deviation from ideality | |
| Bonds (Å) | 0.011 |
| Angles (°) | 1.170 |
| Ramachandran statistics | |
| Favored (%) | 100.0 |
| Disallowed (%) | 0.0 |
| MolProbity clash score | 7.83 |

allowing for measurement of the activity of RasGRP1 variants introduced by transient transfection. Under these conditions, transfection of cells with wild type RasGRP1 leads to ERK phosphorylation in the absence cell stimulation, and this activity is dependent on the RasGRP1 expression level. In addition, the activity of transfected RasGRP1 in unstimulated cells depends on the presence of basal levels of diacylglycerol (*Roose et al., 2007*). Thus, the comparisons of the activities of different RasGRP1 mutants shown below are in the absence of cell stimulation, except where indicated.

The FACS analysis with P-ERK- and Myc-antibodies allowed us to graph P-ERK levels as a function of the Myc-tagged RasGRP1 protein level (*Figure 5D*, *Figure 5—figure supplement 1*). Using this assay, we found that mutation of Asp 453 and Trp 454 in the Cdc25-EF linker increased ERK phosphorylation 1.7 to 2.3-fold, over the entire range of RasGRP1 protein expression levels (*Figure 5D*). Mutation of Trp 454 and two or four other residues in the Cdc25-EF linker led to an additional increase in P-ERK levels. These results confirm that the Cdc25-EF linker plays an inhibitory role in full-length RasGRP1.

## RasGRP1 dimerization is important for autoinhibition

A key step in RasGRP1 activation is thought to be the recruitment of the C1 domain to the membrane by diacylglycerol (*Ebinu et al., 1998*), and the C1 domain of RasGRP1 structurally resembles that of PKCδ, which is shown bound to a diacylglycerol mimic in *Figure 6A*. It is therefore intriguing that the RasGRP1 regulatory module mediates formation of a crystallographic dimer that buries the membrane-binding residues of the two C1 domains (*Figure 6A,B*). The burial of the hydrophobic face of the C1 domain occurs primarily due to contacts with Phe 504, Phe 506 and Val 508 in the EF2 connector and Leu 429 in the Cdc25 domain of the other subunit. Replacement of the hydrophobic connector in the EF2 module with a helix, as is found in most EF hand proteins, would disrupt the shape complementarity of the interface.

Membrane-interacting C1 domains possess several surface-exposed hydrophobic residues that are thought to insert into the lipid membrane upon diacylglycerol recognition (*Zhang et al., 1995*; *Hurley et al., 1997*; *Canagarajah et al., 2004*; *Leonard et al., 2011*). The highly hydrophobic nature of this region favors its sequestration within intramolecular or intermolecular interfaces when the protein is not associated with the membrane. For example, a crystal structure of PKCβ revealed that a C1 domain of that protein clamps onto the kinase domain, thereby preventing nucleotide and diacylglycerol binding (*Leonard et al., 2011*). The formation of the RasGRP1 dimer achieves the same end by burying the membrane-interacting loops of the C1 domain at the dimer interface.

We used cell-based experiments with full-length protein to test the importance of C1-mediated dimerization for RasGRP1 function. These experiments show that disruption of the dimer interface by

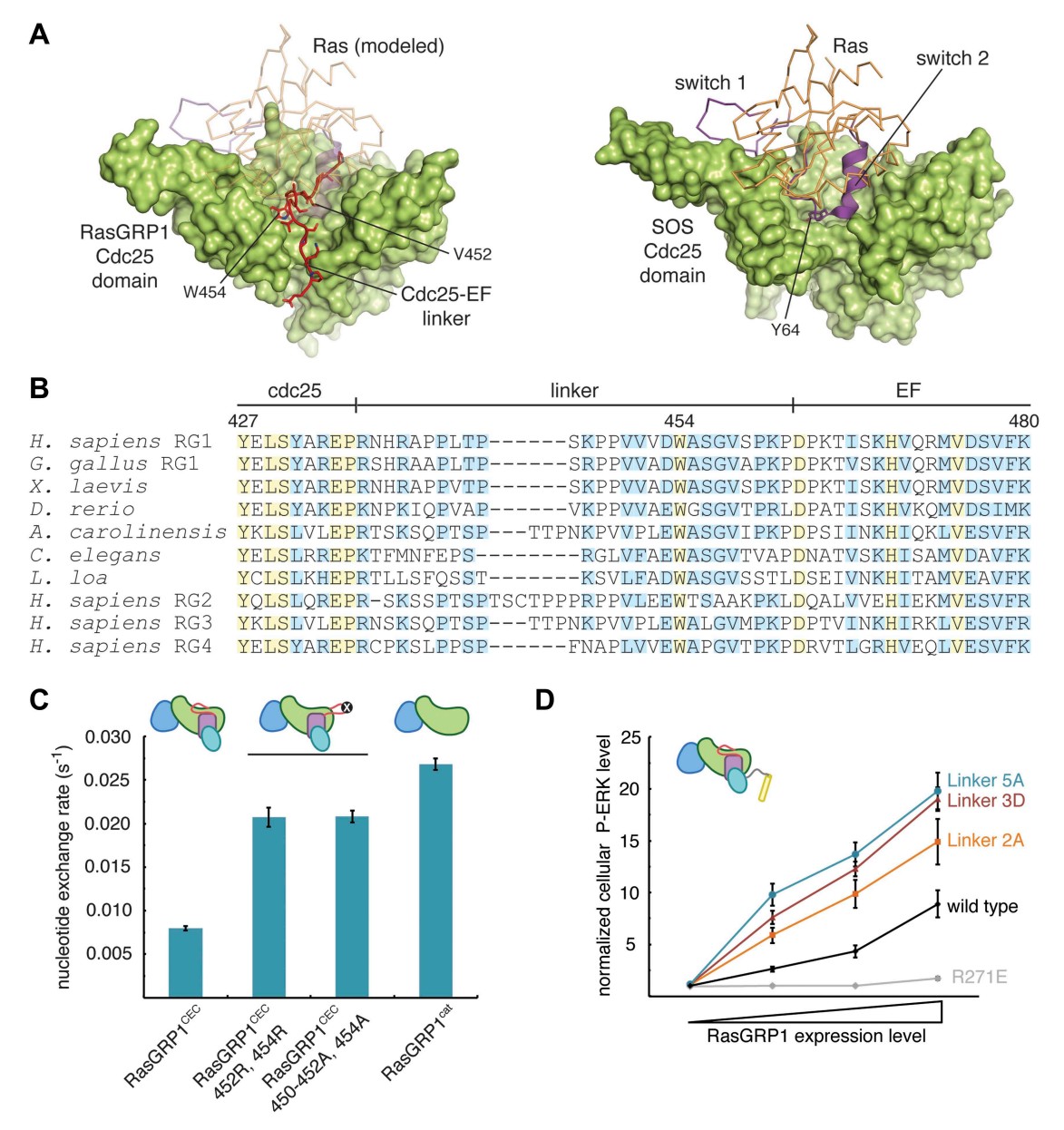

**Figure 5**. The Cdc25-EF linker inhibits RasGRP1. (**A**) The RasGRP1 Cdc25 domain (left) is shown with the Cdc25-EF linker (red) and Ras modeled using the Ras-SOS complex (right). Tyr 64 of switch 2 of Ras makes crucial contacts at the base of the SOS helical hairpin. (**B**) Sequence alignment of the Cdc25-EF linker region of different RasGRP proteins reveals partial (light blue) and complete (yellow) conservation of amino acids important for autoinhibition. (**C**) The in vitro nucleotide exchange activities of different RasGRP1 proteins (10 µM) were compared with 500 nM Ras in solution. Error bars represent ± standard deviation. (**D**) FACS measurements were used to compare ERK phosphorylation in cells expressing full length RasGRP1 (wild type) or mutant proteins with two (Linker 2A), three (Linker 3D) or five mutations (Linker 5A) to the Cdc25-EF linker as a function of expression level. R271E is a catalytically dead mutant that is shown for reference. The average levels are shown with error bars that represent ± SEM.

The following figure supplements are available for figure 5:

**Figure supplement 1**. Cellular analysis of RasGRP1 nucleotide exchange activity.

mutating residues in the EF2 connector that interact with the C1 domain in the crystallographic dimer leads to activation of RasGRP1. Mutation of Phe 506 or Val 508 to Asp resulted in a significant increase in ERK phosphorylation at all expression levels (**Figure 6C**). A similar trend is observed for the Phe 504 Asp protein (data not shown). Mutation of Leu 429 in the Cdc25 domain or Ile 532 in the EF2 module

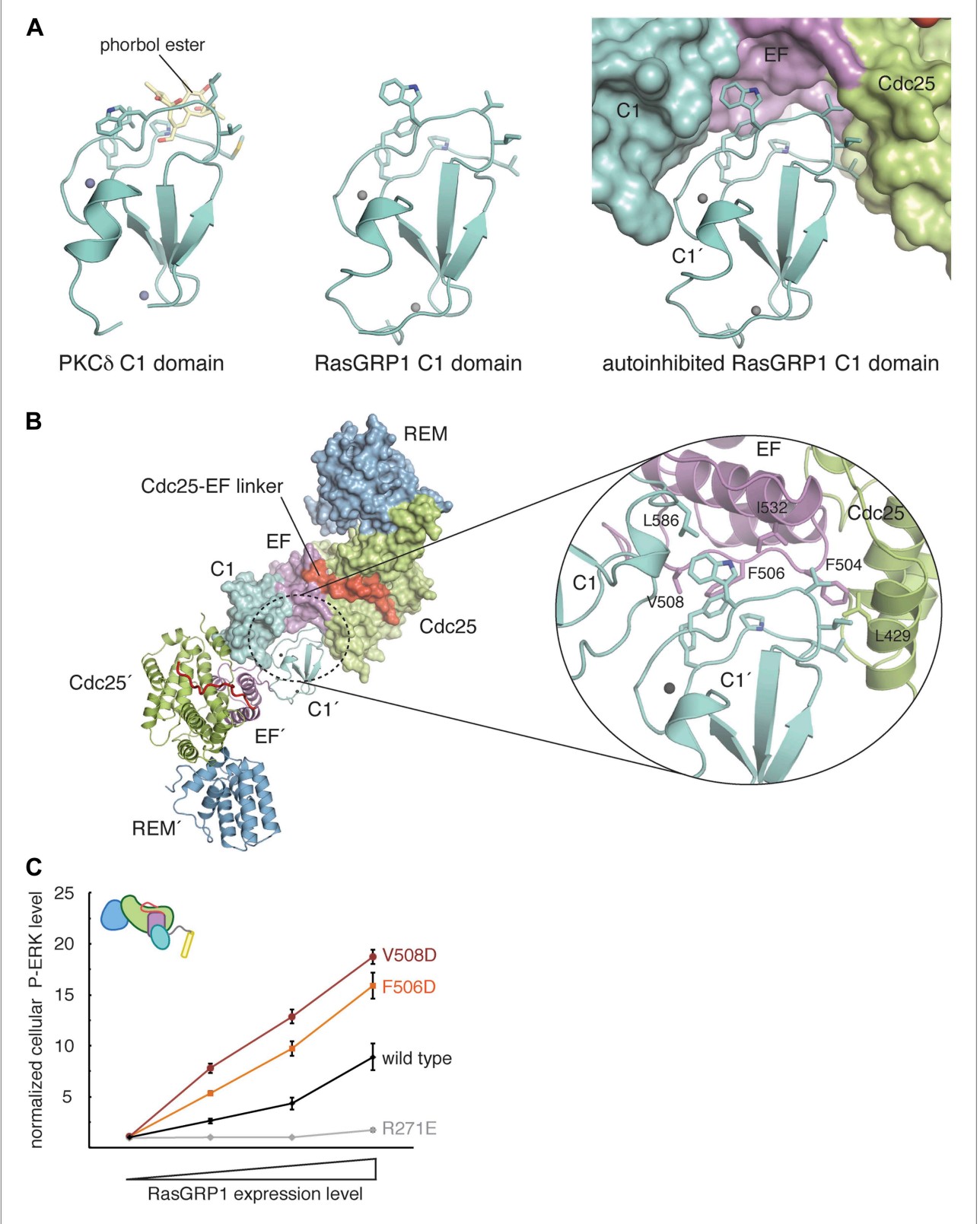

**Figure 6**. RasGRP1 forms an autoinhibited dimer. (**A**) The C1 domain of RasGRP1 (middle) is structurally similar to the C1 domain of PKCδ (PDB ID: 1PTR), shown here bound to a diacylglycerol mimic (phorbol ester) (yellow) (left). The membrane-interacting residues of the C1 domain (teal sticks) are buried at a dimer interface in the RasGRP1 structure (right). (**B**) The dimer interface mediated by the C1 domain buries ~2500 Å² of accessible surface area. The close-up

*Figure 6. Continued on next page*

*Figure 6. Continued*

view of the dimerization surface (right) highlights the importance of the EF2 connector (magenta sticks) as it interacts with the membrane-inserting residues of the C1 domain (teal sticks). (**C**) Dimerization mutants were expressed in Jurkat T cells and P-ERK levels were measured by FACS and compared with wild type RasGRP1. R271E is a catalytically dead mutant that is shown for reference. The average levels are shown with error bars that represent ± SEM.
The following figure supplements are available for figure 6:
**Figure supplement 1**. Oligomerization of RasGRP1CEC.

sharply decreased Ras activation at all expression levels, suggesting that these mutations compromise protein stability (data not shown). Mutation of these residues also prevented soluble RasGRP1 expression in *E. coli*. Overall, the results from mutation of the dimerization interface of RasGRP1 in cells are consistent with the EF2 module playing a crucial role in autoinhibition. We also investigated the oligomeric state of RasGRP1CEC by analytical gel filtration analysis and found that the protein can form dimers and higher order oligomers in vitro (*Figure 6—figure supplement 1*).

The structure of the C-terminal coiled coil demonstrates that a second dimerization interface is present in RasGRP1. Consistent with this structure, RasGRP1CC is exclusively dimeric in solution from 30 μM to 1.5 mM monomer, based on multiangle light scattering measurements (data not shown). The fact that the C-terminal coiled coil is parallel and dimeric is consistent with an ability to readily connect with the autoinhibited catalytic module and reinforce formation of the C1-mediated dimer. Because coiled coil formation is independent of the C1-mediated interface, RasGRP1 is likely to remain dimeric upon disruption of the C1-mediated dimer by membrane binding. Dimerization in this way is expected to enhance the affinity of RasGRP1 for the membrane by allowing the dimer to engage two Ras molecules simultaneously. The possibility that the coiled coil dimer interface plays both positive and negative roles in controlling activity complicates the analysis of mutations in this region, which is still in progress.

## Activation mechanisms of RasGRP1

### The Cdc25 domain of RasGRP1 has intrinsically weak nucleotide exchange activity that is enhanced upon membrane recruitment by the C1 domain

Having established autoinhibition of RasGRP1 by an interdomain linker and dimerization, we next sought to understand how the protein is activated. First, we directly compared the activities of RasGRP1 and SOS by measuring guanine nucleotide exchange rates in vitro using purified catalytic modules. We also compared these rates with the activity of the isolated Cdc25 domain of RasGRF (RasGRFCdc25), which has relatively high intrinsic activity on its own and serves as an important benchmark (*Freedman et al., 2006*). Previous studies have demonstrated that SOS activity is increased substantially when Ras coupled to lipid vesicles is used for a substrate due to the enhanced local concentration of Ras that drives occupancy of the allosteric Ras site (*Gureasko et al., 2008*). We have therefore compared nucleotide exchange rates using Ras in solution and tethered to lipid vesicles.

Based on the open conformation of the RasGRP1 helical hairpin, which is similar to that of RasGRF, we expected RasGRP1cat to possess strong nucleotide exchange activity towards Ras in solution and Ras-coupled vesicles because it is not inhibited by the Cdc25-EF linker. Instead, RasGRP1cat is significantly less active than RasGRFCdc25 with either 500 nM Ras in solution or with Ras coupled to vesicles at the same bulk concentration (*Figure 7A,B*). With both exchange factor and Ras in solution, RasGRP1cat is only fourfold more active than SOScat, which is nearly inactive under these conditions due to low occupancy of the allosteric Ras binding site. Importantly, while SOScat activity increases ~300-fold when measured with Ras-coupled vesicles instead of with Ras in solution, RasGRP1cat activity under these conditions is not enhanced. The weaker activity of RasGRP1cat is consistent with the differences in the activities of the two catalytic modules activating the Ras-ERK pathway in Jurkat cells (*Figure 7C*) and can be rationalized based on the observed differences in the residues that are expected to interact with the switch 1 element of Ras at the active site (*Hall et al., 2001*). Specifically, SOS residues Lys 939 and His 911 are important for Ras recognition, but these residues are replaced by Ser 364 and Asn 336 in RasGRP1.

We tested the effect of localizing Ras to vesicles with different lipid compositions on RasGRP1CEC activity. Unlike the isolated catalytic module, which does not show enhanced activity when Ras is localized

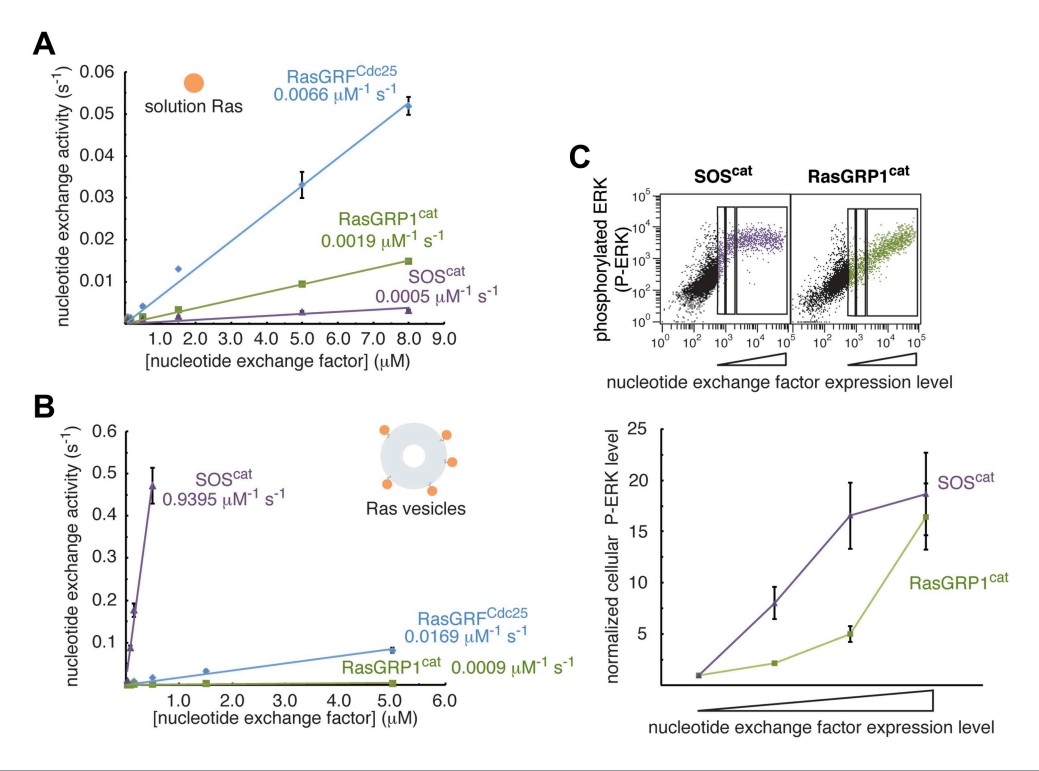

**Figure 7**. RasGRP1 displays relatively weak nucleotide exchange activity. The activities of RasGRF[Cdc25], RasGRP1[cat] and SOS[cat] were measured in vitro with (**A**) Ras in solution (500 nM), or with (**B**) Ras-coupled vesicles using mant-dGDP fluorescence. Error bars represent ± standard deviation. (**C**) Phosphorylated ERK (P-ERK) levels were measured using FACS as a function of nucleotide exchange factor concentration for RasGRP1[cat] and SOS[cat]. Representative dot plots are shown (top) with the gates used for quantitation (bottom). Error bars represent ± SEM.

to vesicles, the presence of the C1 and EF domains in RasGRP1[CEC] results in a sixfold enhancement in activity (with 500 nM RasGRP1[CEC]) on Ras-coupled vesicles containing phosphatidyl choline compared with Ras in solution (*Figure 8*). This activation is likely due to ionic interactions between positively charged residues in RasGRP1, particularly those in the C1 domain, and the negatively charged maleimide-functionalized lipids used for Ras coupling (*Lorenzo et al., 2000*). An additional and very substantial enhancement in activity occurs when 3% diacylglycerol and 10% phosphatidylserine are included in the vesicles, leading to an activation of over 100-fold when compared to reactions with Ras in solution as a substrate. The high Ras density and saturating diacylglycerol levels present in this experiment likely generate the maximal activity possible for this construct. While these conditions are useful for understanding the properties of the system, they are unlikely to occur in a cell. With a constant 5% phosphatidylserine present in the vesicles, increasing the diacylglycerol concentration from 0.4% to 4% increases nucleotide exchange activity ~15-fold (data not shown). We conclude that membrane recruitment by the C1 domain plays a significant role in disrupting the autoinhibitory mechanisms and activating RasGRP1. Previous mutagenesis studies have suggested that RasGRP1 has an allosteric Ras binding site similar to that of SOS (*Tazmini et al., 2009*). However, unlike SOS[cat] activity, which is highly stimulated in the presence of Ras coupled to lipid vesicles, RasGRP1[cat] activity is relatively unchanged. This result indicates that stimulation of RasGRP1 nucleotide exchange activity by Ras binding to an allosteric site is unlikely. We cannot rule out that a GTPase other than Ras can bind to an allosteric site at the interface of the REM and Cdc25 domains.

## The EF domain of RasGRP1 binds a single calcium ion

The sequence of the RasGRP1 EF domain suggests the presence of a pair of EF hands (*Figure 10—figure supplement 1*). Classical Ca2+-binding EF hands, such as calmodulin, contain a pair of helices (the entering and exiting helix) flanking a 12-residue loop that contains acidic residues that form a

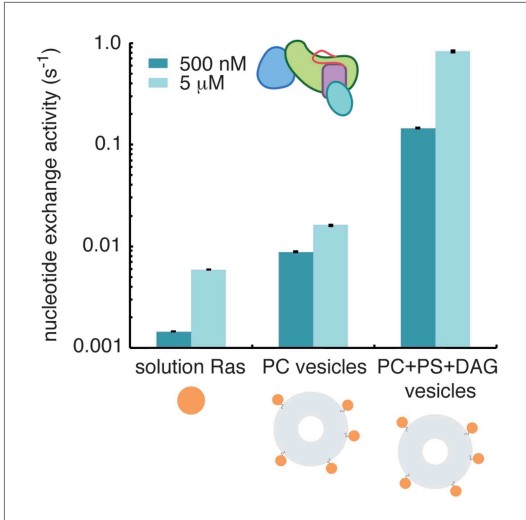

**Figure 8**. Diacylglycerol activates RasGRP1. The activity of RasGRP1[CEC] at 500 nM and 5 µM was measured with Ras in solution, Ras coupled to vesicles containing phosphatidyl choline (PC) or Ras coupled to vesicles with PC with phosphatidyl serine (PS) and diacylglycerol (DAG).

heptagonal bipyramidal coordination sphere around the $Ca^{2+}$ (Gifford et al., 2007). Key residues in the metal-binding loop are identified by number in *Figure 9*.

The presence of a full complement of metal-coordinating residues in both EF hands suggests that the RasGRP1 EF domain can bind calcium. We measured metal binding to the RasGRP1 EF domain using isothermal titration calorimetry (ITC) and NMR. ITC measurements with $Ca^{2+}$ and the isolated EF domain of RasGRP1 (RasGRP1[EF]) are consistent with a single calcium-binding site with a dissociation constant ($K_D$) of 1.3 ± 0.3 µM (*Figure 10B*, *Table 3*). A RasGRP1 construct containing the REM and Cdc25 domains in addition to the EF domain (RasGRP1[CE]) has a fivefold weaker affinity for $Ca^{2+}$ with no change in stoichiometry. The $^1H - ^{15}N$ HSQC spectrum of the isolated EF domain of RasGRP1 with calcium is also consistent with one metal-binding site. There is a single resonance at ~10.5 ppm in the $^1H$ dimension, which is diagnostic of a glycine residue at the sixth position in the loop of metal-bound EF hands because of a strong hydrogen bonding interaction with a metal-coordinating aspartate (*Ikura et al., 1985*) (*Figure 10—figure supplement 2*).

To determine which of the two EF modules interacts with $Ca^{2+}$, potential metal-coordinating residues in each EF loop were replaced by alanine, and the mutant proteins were analyzed for metal binding by ITC and fluorescence spectroscopy. Mutation of the putative metal-binding loop in the EF2 module had little effect on metal-binding stoichiometry. In contrast, mutation of the glutamate in the 12th position in loop of the EF1 module resulted in no detectable interaction with calcium, as measured by ITC (*Table 3*). Similar results were observed for $Tb^{3+}$ binding to RasGRP1[CEC], which was measured using fluorescence resonance energy transfer (see 'Materials and methods') (*Le Clainche et al., 2003*; *Yang et al., 2003*) (*Figure 10—figure supplement 3*). These results demonstrate that RasGRP1 can bind calcium at a site in the EF1 module, but calcium binding to the EF2 module cannot be detected. Presumably, EF2 has lost the ability to strongly bind calcium due to the loss of its entering helix, and this segment has instead evolved into the hydrophobic EF2 connector that contributes to a dimer interface, as outlined above.

RasGRP1 variants bearing mutations in EF1 were tested in Jurkat cells for their ability to activate the ERK pathway upon stimulation with ionomycin, which causes a strong calcium influx into cells. This treatment mimics the increase in intracellular calcium that is triggered by PLCγ through the production of inositol triphosphate. Cells expressing wild type RasGRP1 show three to sixfold activation upon ionomycin treatment relative to untreated cells, depending on the expression level. In contrast, cells that express RasGRP1 EF1 mutants that exhibit decreased metal-binding affinity in vitro are significantly impaired in ERK phosphorylation (25–40% less induction for the lowest expression gate) upon ionomycin treatment (*Figure 10C*). Importantly, the EF1 mutants induce levels of ERK phosphorylation that are similar to wild type RasGRP1 in the absence of ionomycin (data not shown). Attempts to observe the effect of calcium on nucleotide exchange activity in vitro have been unsuccessful due to the presence of high concentrations of $MgCl_2$ and GTP, which are necessary for the in vitro assay but compete with RasGRP1 for $Ca^{2+}$ binding.

## Calcium binding induces a large conformational change in the isolated EF domain

Our measurements show that RasGRP1 binds $Ca^{2+}$ with low micromolar affinity. Because the level of $Ca^{2+}$ in resting cells is in the 20–100 nM range (*Usachev et al., 1995*; *Schwaller, 2010*), this suggests that RasGRP1 will not bind $Ca^{2+}$ until the cell is stimulated and calcium levels rise substantially. We therefore investigated the effects that calcium binding has on the structure of RasGRP1.

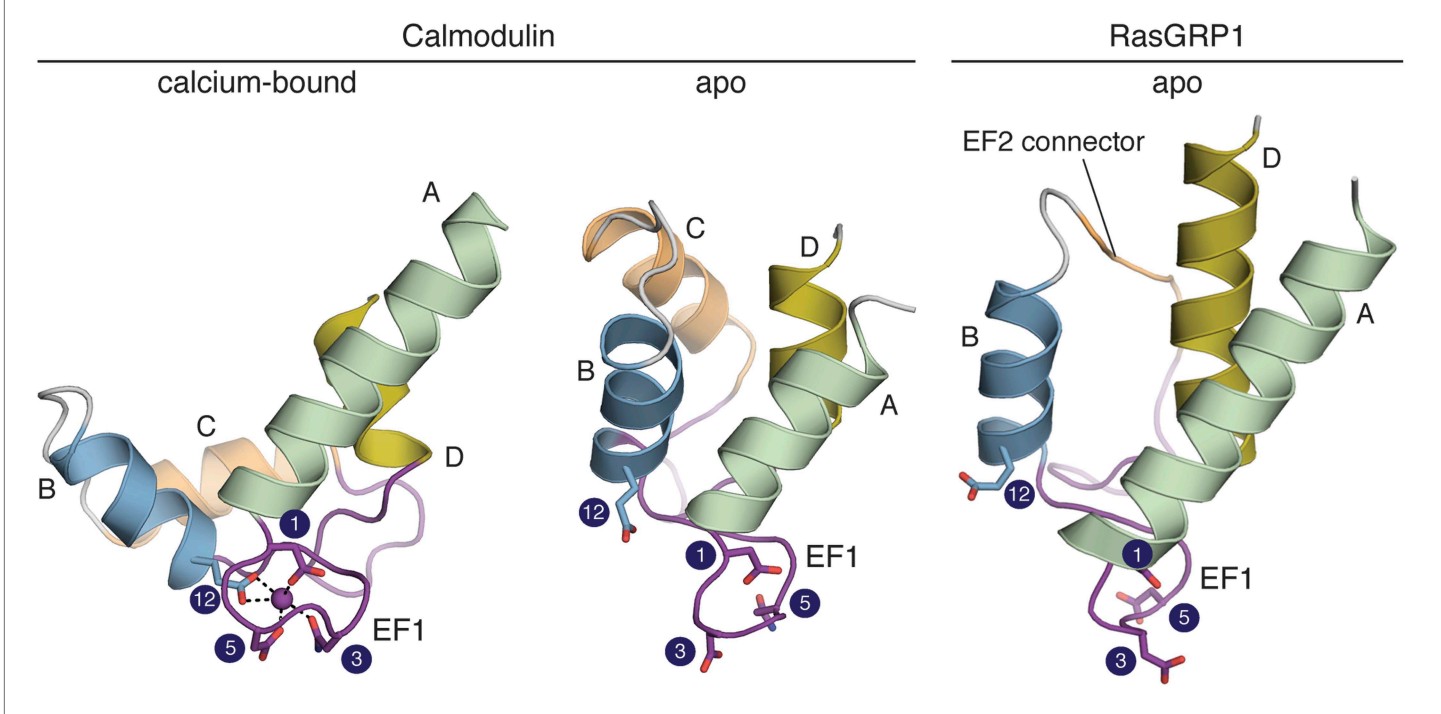

**Figure 9**. The RasGRP1 EF hands adopt a closed conformation. The C-terminal domain of calcium-bound calmodulin (PDB ID: 1CLL) is shown on the left. Helices are denoted by the letters A–D. Comparison to apo calmodulin (PDB ID: 1CFD) (middle) shows that calcium (purple sphere) induces a dramatic rearrangement in the helix orientations for both EF hands. The conformation of the EF domain of RasGRP1 (right) is similar to that of apo calmodulin. The four sidechains that directly contact the $Ca^{2+}$ are shown in sticks. The numbering refers to the positions in the canonical calcium-binding loops as shown in **Figure 10**. The glutamate at position 12 is rotated away from the other metal-binding residues in apo calmodulin and RasGRP1. Unlike other EF-hand pairs, RasGRP1 lacks the entering helix in EF2 (helix C), which is replaced by a short, hydrophobic linker (EF2 connector, orange).

Insight into the effect that calcium binding might have on the RasGRP1 EF domain is provided by the structure of calmodulin. As shown in **Figure 9**, calcium binding induces a change in the orientation of the entering and exiting helices in both EF hands of calmodulin. The EF1 module of RasGRP1 contains the canonical helix-loop-helix structure expected for an EF hand. The angle between helices A and B in the RasGRP1 EF1 module is ~145°, which is similar to the corresponding inter-helix angle in apo-calmodulin (140°) (**Zhang et al., 1995**) but significantly larger than the inter-helix angle in EF1 from the C-terminal domain of $Ca^{2+}$-calmodulin (~100°). In contrast, the EF2 module in RasGRP1 is missing the entering helix (helix C), which is instead replaced by the EF2 connector (**Figure 9**).

A key element in driving the change in the relative orientation of the two helices in each EF hand is the glutamate sidechain at position 12 in the $Ca^{2+}$-chelating loop. In the structure of apo-calmodulin, this sidechain is rotated away from the cluster of other $Ca^{2+}$-coordinating residues by the orientation of helix B. Calcium binding requires a rotation in helix B so that the glutamate sidechain is brought into register with the other $Ca^{2+}$-coordinating residues (compare **Figure 9B** left and middle). EF1 of RasGRP1 is shown in the same orientation in **Figure 9B** (right) and it is obvious that the glutamate sidechain at position 12 is oriented as in apo-calmodulin, pointing away from the calcium-binding loop. The ability of EF1 in RasGRP1 to bind calcium requires that the orientation of helix B undergo a significant adjustment. However, calcium binding induces a range of conformational changes among different EF hand-containing proteins, making it difficult to model the conformational change in RasGRP1 with confidence.

We investigated the structural effects of calcium binding to RasGRP1$^{EF}$ using NMR. The $^1H$ - $^{15}N$ HSQC spectrum of metal-free RasGRP1$^{EF}$ shows very broad resonances with poor chemical shift dispersion, consistent with a relatively dynamic protein and similar to many other metal-free EF-hand proteins (**Ames et al., 1999**; **Veeraraghavan et al., 2002**; **Aravind et al., 2008**). Circular dichroism spectra revealed significant helicity for the protein in the apo state, indicating that it is not unfolded (**Figure 10—figure supplement 4**). Calcium addition resulted in dramatic changes to the RasGRP1$^{EF}$

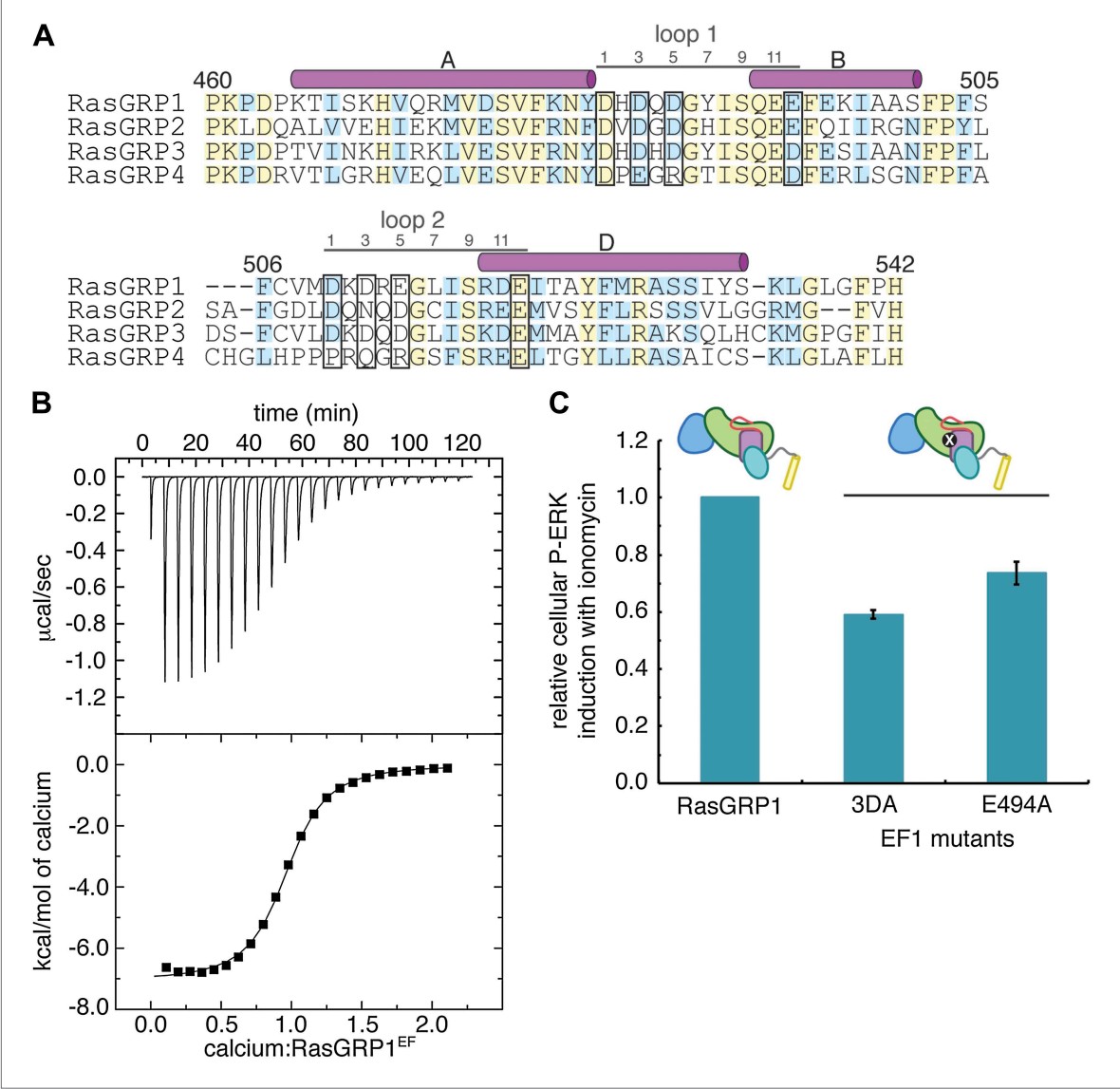

**Figure 10**. Calcium binds to the EF domain of RasGRP1. (**A**) Sequence alignment of the four human RasGRP proteins shows differences in the residues that directly contact metal ions in canonical EF hands (black boxes) and in the number of residues between the two loops. Numbering refers to human RasGRP1. (**B**) A representative ITC curve from 700 μM CaCl₂ titrated into 50 μM RasGRP1$^{EF}$ at 20°C is shown with the baseline corrected raw data (top). The integrated heats of interaction are fit to a one set of sites model (bottom) with average fitting parameters listed in **Table 3**. (**C**) Relative P-ERK levels are shown for wild type RasGRP1 and proteins with mutations to EF1 after stimulation with ionomycin. The 3DA mutant contains alanine mutations at the 1, 3 and 5 positions in the calcium-binding loop of EF1. The E494A mutant contains an alanine at position 12 in the calcium-binding loop of EF1. Measurements for the mutants were normalized to levels for the same construct without ionomycin and plotted as the fraction of activation compared to WT for the lowest RasGRP1 expression gate. Similar results were obtained at higher expression levels, but are omitted for clarity. The average levels are shown with error bars that represent ± SEM.

The following figure supplements are available for figure 10:

**Figure supplement 1**. Sequence alignment of EF-hand proteins.

**Figure supplement 2**. RasGRP1$^{EF}$ undergoes a significant conformational change upon calcium binding.

**Figure supplement 3**. Metal ions bind to the EF1 module of RasGRP1.

*Figure 10. Continued on next page*

*Figure 10. Continued*

**Figure supplement 4**. Circular dichroism of RasGRP1$^{EF}$.

**Figure supplement 5**. Calcium binding to RasGRP2$^{EF}$ induces a significant conformational change.

**Figure supplement 6**. RasGRP2$^{EF}$binds two calcium ions.

spectrum, although it is still plagued by line broadening and overlapping resonances, which could be caused by conformational heterogeneity of the domain. Evaluation of $^1$H - $^{15}$N HSQC spectra using different protein constructs at various temperatures, pH and with detergents led to a slight improvement of the data quality but prevented a full structural analysis of the calcium-bound form of RasGRP1$^{EF}$. Nonetheless, we conclude that calcium induces significant changes in conformation and/or dynamics of the EF domain.

As an alternative to studying Ca-RasGRP1$^{EF}$, we used NMR to analyze the EF domain of RasGRP2 (RasGRP2$^{EF}$), which shares 47% sequence identity with RasGRP1$^{EF}$ (***Figure 10A***). While the NMR spectrum for apo-RasGRP2$^{EF}$ shows poor chemical dispersion (similar to that of apo-RasGRP1$^{EF}$), the calcium-bound form of the protein is well behaved with the expected ~75 well-resolved backbone resonances, and the spectrum suggests that a significant conformational change occurs in the protein upon calcium binding (***Figure 10—figure supplement 5***). The circular dichroism spectra for apo- and Ca-RasGRP2$^{EF}$ are similar to those of RasGRP1$^{EF}$, with indications of helical content observed in both states (***Figure 10—figure supplement 4***). The HSQC spectrum for RasGRP2$^{EF}$ showed two peaks at ~10.5 ppm in the $^1$H dimension, which are indicative of two calcium binding sites (***Ikura et al., 1985***), rather than the single site seen in RasGRP1. Calcium titrations using ITC are consistent with this observation ($N = 1.8 \pm 0.1$ $K_D = 80 \pm 18$ nM at 20°C) (***Figure 10—figure supplement 6***).

The high quality of the NMR spectra for RasGRP2$^{EF}$ allowed us to determine the structure of the EF domain bound to Ca$^{2+}$. We calculated a structural ensemble using Nuclear Overhauser effect (NOE) distance restraints and further refined the ensemble using residual dipolar couplings (RDCs) to more accurately orient the helices of RasGRP2$^{EF}$ (***Figure 11—figure supplement 1***, ***Table 4***). The backbones of the two metal-binding loops adopt conformations capable of metal coordination prior to any calcium-imposed restraints, consistent with both binding sites being occupied. The two extra residues in the helix C region of RasGRP2 relative to RasGRP1 (***Figure 10A***) allow the formation of a short helical turn in EF2, which enables the second metal-binding site to be formed. Comparison of apo-RasGRP1$^{EF}$ and Ca$^{2+}$-RasGRP2$^{EF}$ shows a significant rearrangement in the helical orientation in EF1 with the angle between helices A and B opening by ~40° upon calcium binding, similar to the change observed for calmodulin (***Figure 11***, ***Figure 11—figure supplement 2***). Given the high sequence similarity between the EF domains of the two RasGRP proteins, particularly in EF1, it is reasonable to assume that the EF1 module of RasGRP1 will undergo a similar structural change upon binding to a calcium ion.

## Implications of Ca$^{2+}$ binding for release of RasGRP1 autoinhibition

The EF domain of RasGRP1 docks onto the Cdc25 domain using the A and B helices of EF1 (***Figure 11***). To illustrate the potential consequences of Ca$^{2+}$ binding to the EF1 module, we docked the Ca$^{2+}$-RasGRP2 EF domain onto the apo-EF1 module in the crystal structure of RasGRP1 by superimposing helix B. We aligned the B helices because the presence of conserved bulky

**Table 3.** Summary of ITC data for Ca$^{2+}$ binding to RasGRP1

| | *N* | Δ*H* (kcal/mol) | $K_D$ (μM) | T (°C) |
|---|---|---|---|---|
| RasGRP1$^{EF}$ | 0.9 ± 0.1 | −7.5 ± 0.5 | 1.3 ± 0.3 | 20 |
| RasGRP1$^{EF}$ | 1.0 ± 0.1 | −5.0 ± 0.2 | 0.75 ± 0.10 | 15 |
| RasGRP1$^{CE}$ | 1.0 ± 0.3 | −5.9 ± 0.4 | 3.7 ± 0.3 | 15 |
| RasGRP1$^{CE}$ E494A | no binding | | | 15, 20 |

**Table 4.** Statistics for 10 lowest energy structures of Ca²⁺-RasGRP2ᴱᶠ

| Type of restraint | Number of restraints | Violations per structure |
|---|---|---|
| Total NOEs | 653 | 0 (> 0.5 Å) |
| Intraresidue | 221 | 0 |
| Sequential NOEs | 299 | 0 |
| Medium-range NOEs | 78 | 0 |
| Long-range NOEs* | 55 | 0 |
| R.m.s deviations from experimental distance restraints (Å) | | 0.044 ± 0.0005 |
| Hydrogen bond restraints | 27 | |
| Dihedral angle restraints (φ, ψ and χ¹)† | 137 | 0 (>5°) |
| R.m.s deviations from experimental dihedral restraints (°) | | 0.503 ± 0.1219 |
| Dipolar coupling restraints (Hz)‡ | | |
| NH | 68 | 0.85 ± 0.04 |
| Cα-Hα | 29 | 3.15 ± 0.07 |

**Structure quality factor – overall statistics§**

| | Mean score | SD | Z-score |
|---|---|---|---|
| Procheck G-factor (phi/psi only) | 0.07 | N/A | 0.59 |
| Procheck G-factor (all dihedral angles) | −0.29 | N/A | −1.71 |
| Verify3D | 0.20 | 0.0424 | −4.17 |
| Prosall (-ve) | 0.28 | 0.0395 | −1.53 |
| MolProbity clashscore | 30.77 | 3.6352 | −3.75 |

**Deviation from idealized covalent geometry**

| | |
|---|---|
| Bonds (Å) | 0.0019 ± 0.00003 |
| Angles (°) | 0.3178 ± 0.0013 |
| Impropers (°) | 0.4386 ± 0.0196 |

**Average pairwise RMSD (Å)§**

| | |
|---|---|
| All heavy atoms | 1.0 |
| Backbone heavy atoms | 0.5 |

**Ramachandran plot statistics (%)#**

| | |
|---|---|
| Most favored region | 93.4 |
| Additionally allowed region | 6.6 |

*Sidechains coordinating the Ca²⁺ ion in EF hands 1 and 2 were implemented as distance restraints.
†χ¹ angles were obtained from PREDITOR (**Berjanskii et al., 2006**), and φ and ψ angles were calculated using TALOS+ (**Shen et al., 2009**).
‡The r.m.s differences (Hz) between the experimental RDC data and back-calculated RDCs from the 10 lowest energy structures were obtained by SVD fitting using the program PALES (**Zweckstetter and Bax, 2000**).
§The pairwise RMSD for residues 421-486 and structure quality factor were generated using the Protein Structure Validation Software suite v1.4 (**Bhattacharya et al., 2007**).
#The Ramachandran statistics were evaluated for residues 422-486 with PROCHECK (**Laskowski et al., 1996**).

hydrophobic residues at the interface with the Cdc25 domain suggests that helix B may be docked stably onto the Cdc25 domain (**Figure 11**, left). In contrast, the entirely polar interface between helix A and the Cdc25 domain makes it more likely that helix A will be dislodged upon Ca²⁺ binding.

Assuming that helix B remains docked onto the Cdc25 domain, the structure of the Ca²⁺-bound EF domain of RasGRP2 suggests that helix A will rotate outwards by ~40° upon calcium binding (**Figure 11**). This would completely disrupt the interface between the EF domain and the two C1 domains of the dimer. Note that this conclusion is not dependent on the docking alignment because

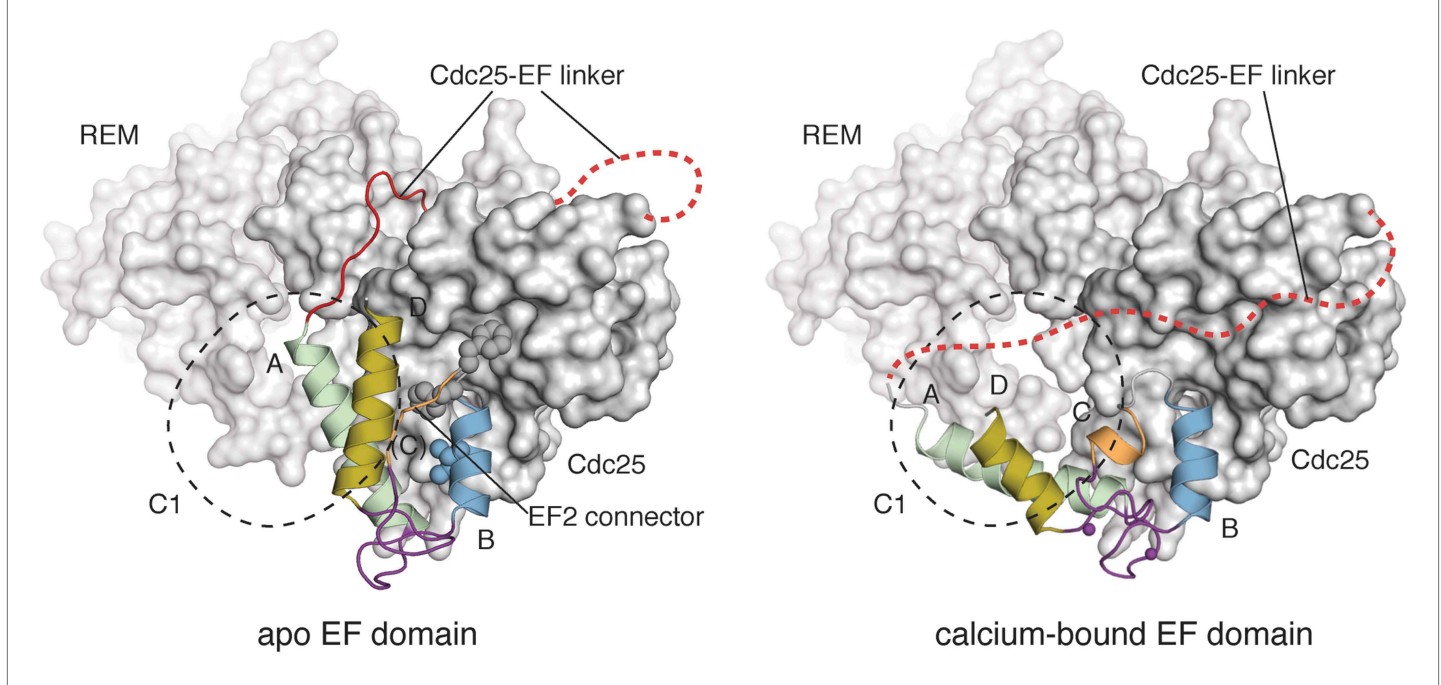

**Figure 11**. Calcium binding induces a large conformational change in the EF domain. The helices of the EF domains of apo RasGRP1 (left) and calcium-bound RasGRP2 (right) are shown with the Cdc25 domain of RasGRP1. The two EF domains were superimposed using helix B, which contains three hydrophobic residues (spheres) that interact extensively with the Cdc25 domain. Using this frame of reference, the angle between helices A and B changes by ~40° upon calcium binding. The conformational change in the EF domain could disrupt the docking of the Cdc25-EF linker (red). The footprint of the C1 domain is shown with a dotted black line.

The following figure supplements are available for figure 11:

**Figure supplement 1**. NMR analysis of Ca RasGRP2[EF].

**Figure supplement 2**. Comparison of apo RasGRP1[EF] and Ca RasGRP2[EF].

of the intimate contact between the EF domain and two C1 domains in the dimer. The tight packing between the two C1 and EF domains must be dismantled to accommodate the change in angle between the A and B helices of EF1.

Because of the direct connection between helix A and the Cdc25-EF linker, it is also possible that the movement of helix A upon calcium binding would dislodge the Cdc25-EF linker from the Ras binding site. Based on the alignment using helix B, the tip of helix A moves away from the Cdc25 domain by ~18 Å, which would require the Cdc25-EF linker to follow a path along the periphery of the Cdc25 domain rather than traversing the Ras binding site. Additional mechanisms may be necessary to completely remove this segment from the Ras-binding site. For example, Cdc25 domain phosphorylation (*Limnander et al., 2011*) near the docking surface for the Cdc25-EF linker could weaken the inhibitory interaction. The binding of other proteins to the proline-rich segment of the linker may also influence autoinhibition.

Comparison of the two structural models shown in *Figure 11* makes it clear that the presence of a second Ca²⁺-binding site in RasGRP2 does not alter the essence of this argument. The second Ca²⁺-binding site is accommodated by an additional two residues in the region corresponding to helix C, allowing for the formation of a helical turn that is absent in RasGRP1. The helix C region and helix D are located on the outer surface of the EF domain, away from the Cdc25 domain. The interactions with the C1 domains will likely be different in RasGRP2, but the structural change involving EF1 is likely to be conserved. EF-hand pairs with a single calcium-binding site have been observed in other proteins such as the polycystin-2 ion channel, which also undergoes a significant conformational change in response to calcium (*Petri et al., 2010*).

## Concluding remarks

Our structural and functional studies on the autoinhibition and activation of RasGRP1 reveal striking contrasts between the control of RasGRP1 and SOS, reflecting how the two exchange factors respond differently to thymocyte T cell receptor inputs. RasGRP1 is maintained in an autoinhibited, dimeric state, which is released by the coordinated binding of diacylglycerol and calcium second messenger molecules (*Figure 12*). In contrast, SOS employs a conformational switch in the helical hairpin that is coupled to the allosteric Ras-binding site (*Margarit et al., 2003*; *Sondermann et al., 2004*; *Gureasko et al., 2008*, *2010*). The N-terminal membrane-binding domains of SOS are coupled to this switch because they physically block access to the allosteric Ras-binding site. Another distinction between RasGRP1 and SOS is that dimeric RasGRP1 has buried membrane-interacting surfaces whereas the membrane binding sites in SOS appear to be readily accessible. SOS may not require masking of its membrane interacting surfaces because its activation requires occupation of the allosteric Ras-binding site. In contrast, RasGRP1 appears competent for Ras activation as soon as autoinhibitory restraints are released. These differences in regulatory mechanisms underlie ultrasensitive signaling by SOS, while the activation of Ras signaling by RasGRP1 occurs in a graded manner in response to stimulation. Further experiments will be necessary to understand how the different signaling inputs such as diacylglcyerol, calcium and phosphorylation are integrated by RasGRP1 to produce this signaling pattern.

Our finding that the Cdc25-EF linker physically blocks the Ras binding site in RasGRP1 fits into a theme for regulation of exchange factors by steric blockage of the small GTPase binding site. For example, Sec7 domain-containing Arf exchange factors are autoinhibited by an interdomain linker and a regulatory helix (*DiNitto et al., 2007*). Similarly, the regulatory domains of Epac block the Rap binding site, and cAMP binding releases autoinhibition (*Rehmann et al., 2006*, *2008*). SOS is an exception to this theme, although it is possible that the C-terminal segment of SOS, for which no structural information is available but which is known to inhibit nucleotide exchange activity (*Aronheim et al., 1994*; *Wang et al., 1995*), may play a similar role.

*C. elegans* possess a single RasGRP protein, whereas mammals have four RasGRP proteins with distinct but often overlapping tissue distributions. It appears that these four proteins have evolved

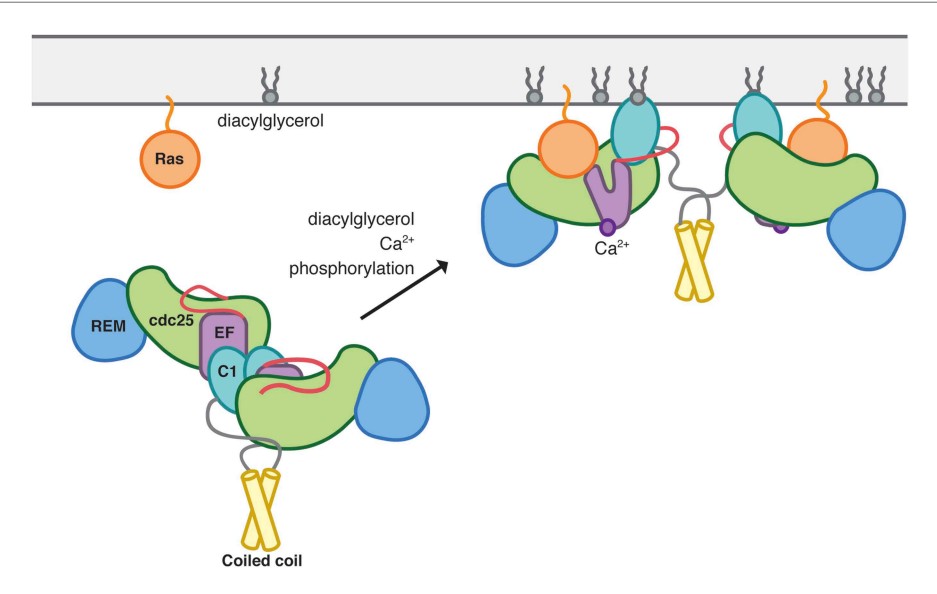

**Figure 12**. Model of RasGRP1 activation. Inactive RasGRP1 (left) is stabilized by the C1-dimer interface, which sequesters the membrane-interacting surface of the C1 domain, and the active-site blocking Cdc25-EF linker (red). The C-terminal coiled coil stabilizes the dimer, thereby preventing inappropriate Ras activation. The autoinhibited form is activated by multiple signaling inputs that enhance nucleotide exchange activity (right). Diaclyglycerol binding disrupts C1 dimerization, while $Ca^{2+}$ binding to EF1 causes a conformational change that contributes to C1 reorientation, and the release of the inhibitory segment from the Ras-binding surface. Phosphorylation of the Cdc25 domain could aid in removal of the inhibitory linker.

partially overlapping regulatory mechanisms. While the hydrophobic portion of the Cdc25-EF linker that docks onto the Cdc25 domain is conserved among the four human RasGRP proteins, only RasGRP1 contains a C-terminal coiled coil. This suggests that the Cdc25-EF linker is inhibitory for each protein, but indicates that RasGRP2, 3 and 4 are either not dimeric upon activation or utilize distinct oligomerization mechanisms. The calcium-binding properties among the proteins of this family are also different. For example, examination of the EF domain sequence of RasGRP4 suggests that this protein does not bind calcium, or does so in a manner distinct from classical EF hands. It is perhaps due to these distinctions that loss of RasGRP1 is so detrimental to thymocyte development (*Dower et al., 2000*) and only minimally compensated for by RasGRP3 or 4 (*Zhu et al., 2012*; *Golec et al., 2013*). *RasGRP1* also selectively appears as a common integration site in murine leukemia virus screens (*Mikkers et al., 2002*; *Suzuki et al., 2002*; *Akagi et al., 2004*), resulting in increased expression of RasGRP1, which is not observed for other Ras-specific exchange factors (*Hartzell et al., 2013*).

Nucleotide exchange factors have evolved distinct solutions to tightly tune Ras activation in response to diverse external stimuli. Germline gain-of-function *SOS* alleles can perturb Ras signaling, leading to Noonan syndrome, and the structures of SOS have aided the functional analysis of these variants and have further enhanced our understanding of SOS regulation (*Roberts et al., 2007*; *Tartaglia et al., 2007*; *Findlay et al., 2013*). We anticipate that the mechanisms of RasGRP1 that we have identified will likewise provide a structural framework for understanding the connections between RasGRP1 variants and diseases such as leukemia and systemic lupus erythematosus.

## Materials and methods

### Molecular biology

The genes for human RasGRP1$^{CEC}$ (residues 50–607), RasGRP1$^{EF}$ (residues 459–540) RasGRP2$^{EF}$ (residues 417–495) and RasGRP1$^{CC}$ (residues 739–793) were cloned into the pSMT3 vector, containing N-terminal 6xHis and Sumo tags, at the BamHI and XhoI restriction sites. RasGRP1$^{cat}$ (residues 50–468) was cloned into a pET28 derivative with a tobacco etch virus (TEV) protease-cleavable C-terminal 6xHis tag at the NdeI and XhoI sites. RasGRP1 and SOS mammalian expression plasmids for flow cytometry assays were constructed using the pEF6 vector as described previously (*Roose et al., 2005*, *2007*). Mutations were introduced by Quikchange site-directed mutagenesis (Agilent Technologies, Santa Clara, CA).

### Protein expression and purification

RasGRP1 proteins were expressed in *E. coli* BL21(DE3) cells grown at 37°C in Terrific Broth with 50 µg/ml kanamycin grown to an OD of ~1.0. Cells were then induced with 1 mM IPTG at 15°C (RasGRP1$^{CEC}$, RasGRP1$^{EF}$, RasGRP2$^{EF}$, RasGRP1$^{CC}$) or 18°C (RasGRP1$^{cat}$) for 14–18 hr. Growth media for cells expressing RasGRP1$^{CEC}$ was supplemented with 30 µM ZnCl$_2$. SeMet-substituted protein was expressed in cells grown in M9 minimal media with 50 µg/ml kanamycin. Before induction, the growth media was supplemented with 50 µg/ml of each amino acid (except Met), 5 µg/ml Met and 50 µg/ml SeMet. $^{15}$N, $^{13}$C-labeled proteins were expressed in M9 minimal media with 1 g/L $^{15}$N ammonium chloride and 3 g/L $^{13}$C glucose (Cambridge Isotope Laboratories, Andover, MA).

Cell pellets were resuspended in Buffer A (25 mM Tris [pH 8.0], 500 mM NaCl, 10% glycerol, 20 mM imidazole and 5 mM β-mercaptoethanol [βME]) and frozen at −80°C. All purification steps were carried out at 4°C using columns from GE Healthcare (Piscataway, NJ). Cells were lysed with a cell disrupter with 5 mM βME, 200 µM AEBSF, 5 µM leupeptin and 500 µM benzamidine. Clarified lysates were applied to a 5 ml HisTrap FF affinity column equilibrated in Buffer A. The column was then washed with 100 ml Buffer A, and proteins were eluted in Buffer A with 500 mM imidazole. Proteins were immediately desalted in Buffer B (25 mM Tris, 100 mM NaCl, 10% glycerol and 1 mM TCEP) using a HiPrep 26/10 desalting column. For RasGRP1$^{CEC}$ and RasGRP1$^{cat}$ the buffer pH was 8.5, while the pH for the buffer for all other constructs was 8.0. Purification tags were removed by addition of ULP1 protease (6xHis-Sumo tag) or TEV protease (C-terminal 6xHis-tag) and incubation at 4°C for 14–18 hr. Proteases and tags were removed by a second pass over a HisTrap FF column in Buffer A. The flow-through and wash fractions were concentrated to 2 ml and further purified on a Superdex 26/60 column equilibrated in Buffer B. Fractions containing pure protein were concentrated and frozen at −80°C until use. Ras, SOS and RasGRF constructs were expressed and purified as described previously (*Freedman et al., 2006*;

*Gureasko et al., 2008*). Purification of RasGRP1[CEC] and RasGRP1[cat] yielded ~1 mg of protein per liter of cells, while the other constructs yielded 5–10 mg of protein per liter of cells.

## Crystallization and structure determination

Crystallization of RasGRP1[CEC] was carried out initially with sparse matrix screening using a Phoenix crystallization robot (Art Robbins Instruments, Sunnyvale, CA), and thin hexagonal rod-shaped crystals were obtained in a single condition. The initial hit was further optimized through additive screening. Crystals used for data collection were grown by hanging drop vapor diffusion (500 µl reservoir volume) by mixing 1 µl of protein (10 mg/ml) with 1 µl of 0.15 M sodium citrate tribasic, 22% PEG 3350 and 1 mM $MnCl_2$. Crystals appeared at 20°C in 1–2 days and grew to a maximum length of ~200 µm over 3–5 days. Crystals were cryoprotected in the crystallization solution with 20% glycerol and flash frozen in liquid nitrogen.

RasGRP1[CC] (10 mg/ml) was crystallized in 20 mM sodium acetate (pH 3.6), 22% PEG 3350, 100 mM lithium sulfate and 0.4% formamide by mixing 0.2 µl protein with 0.2 µl of well solution. Square plate-like crystals were harvested after 5–7 days and cryoprotected in the crystallization solution with 20% glycerol. Diffraction data for both RasGRP1[CEC] and RasGRP1[CC] were collected at 100 K on beamline 8.2.2 at the Advanced Light Source, Lawrence Berkeley National Laboratories.

X-ray data were processed with XDS (*Kabsch, 2010*), then Pointless and Scala from the CCP4 program suite (*Winn et al., 2011*). Refinement was performed with Phenix.refine (*Adams et al., 2010*). For RasGRP1[CEC], an initial molecular replacement solution was found using Phaser (*McCoy et al., 2007*) with the RasGRF Cdc25 domain and the core of the SOS REM domain. The location of the C1 domain was identified from anomalous data from the two intrinsic $Zn^{2+}$ ions and the proper orientation was defined by incremental rotation about the axis defining the two metal ions and refinement of the resulting structures. The correct sequence register was determined through identification of 14 of the expected 15 selenium sites using X-ray data for the SeMet-substituted protein (the anomalous peak for residue Met 50 was not present). The position of the Cdc25-EF linker was determined from averaged kick omit maps (*Praznikar et al., 2009*) generated in Phenix, which aid in removing model bias. The RasGRP1[CC] structure was solved by molecular replacement using the APC coiled coil (*Day and Alber, 2000*).

The structural model for RasGRP1[CEC] spans residues 53–593 and includes the REM, Cdc25, EF and C1 domains. The electron density for portions of the linkers between the REM and Cdc25 domains (residues 186–192) and between the Cdc25 and EF domains (437–448) is poor and therefore these residues have been excluded from the final model. The model for RasGRP1[CC] contains two molecules in the asymmetric unit that form the functional unit. Molecule A contains residues 745–793, while molecule B includes residues 745–786.

## In vitro nucleotide exchange assay

Ras-coupled vesicles for in vitro nucleotide exchange experiments were generated as previously described (*Gureasko et al., 2008*). Experiments with Ras-coupled vesicles were repeated at different Ras densities, which ranged from ~1500 to 7000 Ras/µm². Nucleotide exchange measurements were measured by mant-dGDP (Jena Bioscience, Jena, Germany) fluorescence using a stopped-flow apparatus with a Fluoromax-3 fluorometer (Horiba Scientific, Edison, NJ) and analyzed as described (*Gureasko et al., 2008*). The final Ras concentration was 500 nM. Samples were excited at 370 nm (5 nm slit width) and the emission at 430 nm (5 nm slit width) was followed for at least eight minutes with a 0.5 s sampling interval.

## Flow cytometry assays

DNA was introduced into JPRM441 cells (*Roose et al., 2005*) by electroporation ($20 \times 10^6$ cells with 20 µg DNA) (Biorad Genepulser Xcel). After recovery, cells were washed in RPMI, plated in a 96 well round-bottom plate ($0.4 \times 10^6$ cells /well), starved, and stimulated for 5 min with DMSO or with 1 µM ionomycin. Cells were fixed with prewarmed (37°C) Fixation Buffer (BD Cytofix, BD Biosciences, San Jose, CA) and permeabilized with methanol at −20°C. Barcoding protocols were modified from described methods (*Krutzik and Nolan, 2006*). Pacific Blue and Alexa Fluor 488 carboxylic acid succinimidyl-esters (Life Technologies, Grand Island, NY) were added in methanol in serial dilutions, and cells were incubated for 30 min at −20°C. Cells were washed thoroughly in PBS containing 1% BSA and 2 mM EDTA (FACS buffer), and barcoded cells were pooled and incubated with anti-phospho-ERK (clone 197G2) and anti-myc (clone 9B11; Cell Signaling, Danvers, MA) for 1 hr at 22°C. Cells were washed in FACS buffer and

incubated with allophycocyanin-Donkey anti-Rabbit IgG (Jackson ImmunoResearch, West Grove, PA) and Pe/Cy7 Goat anti-mouse IgG (Biolegend, San Diego, CA) for 30 min at 22°C. Cells were washed and analyzed using the LSR II flow cytometer (BD Biosciences, San Jose, CA). Data were analyzed using Cytobank. The specific RasGRP1 mutations made for the Cdc25-EF linker variants are: Linker 2A- W454A and D453A, Linker 3D- V451D, V452D and W454D, Linker 5A- V450A, V451A, V452A, D453A and W454A.

## Calcium binding assays

ITC experiments were carried out using a VP-ITC instrument (GE Healthcare) and analyzed using the Origin 7 software package. RasGRP proteins were treated with 10- to 20-fold excess EDTA for at least 12 hr at 4°C to remove bound metal ions. The sample was then dialyzed extensively against Chelex-treated (Sigma Aldrich, St. Louis, MO) 25 mM Tris (pH 8.0 or 8.5), 100 mM NaCl, 10% glycerol and 1 mM TCEP for 15 hr to remove the EDTA. Protein (20–50 μM) was placed in the cell, and aliquots of 0.2–1.0 mM $CaCl_2$ were titrated into the protein sample. An initial injection of 4 μl was excluded from data analysis, followed by 23 injections of 12 μl each, separated by 300 s with a filter period of 2 s. Titration curves were fit with a "one set of sites" model, with the fitting parameters $N$ (stoichiometry), $K_a$ (association constant) and $\Delta H$ (enthalpy of interaction).

Calcium binding to RasGRP1[CEC] could not be analyzed by ITC due to poor stability of the protein during stirring, and difficulties removing metal ions pre-bound to the EF domain without removing $Zn^{2+}$ from the C1 domain. Instead, metal binding was analyzed using a $Tb^{3+}$-FRET assay by excitement of tryptophan and tyrosine residues, which can transfer energy to nearby $Tb^{3+}$ ions bound to the protein. Protein (1 μM) was incubated at 22°C for 30 min with varying $Tb^{3+}$ concentrations, and the fluorescence emission at 543 nm was recorded after excitation at 285 nm. A separate sample was made for each point of the titration. Data were fit to a single binding site isotherm.

## NMR

Isotopically labeled proteins for NMR were prepared as described above. For RasGRP2[EF] samples, 500 μM protein was mixed with 1.2 mM $CaCl_2$ or 5 mM EDTA in 25 mM Hepes (pH 7.0), 100 mM NaCl, 1 mM TCEP and 7% $D_2O$. RasGRP1[EF] (250 μM) samples were made in 25 mM Hepes (pH 7.0), 100 mM NaCl, 1 mM TCEP, 15 mM β-octyl glucoside and 7% $D_2O$ with 400 μM $CaCl_2$ or 5 mM EDTA.

All NMR experiments were recorded on a Bruker Avance II 800 MHz NMR spectrometer (Bruker Biospin Corp, Billerica, MA) equipped with a room temperature TXI probe at 304 K. Data were processed with NMRPipe (*Delaglio et al., 1995*; *Goddard and Kneller*) and analyzed using Sparky 3.111 (*Goddard and Kneller*). The cross-peak fit height was measured using the Gaussian line fitting protocol implemented in Sparky 3.111. The 2D $^{15}N$-$^1H$-HSQC spectra were acquired with spectral widths of 35 and 14 ppm for the $^{15}N$ and $^1H$ dimensions, respectively and with 128 ($^{15}N$) and 1024 ($^1H$) complex points. During all experiments the carrier frequencies of the proton and nitrogen channels were centered at 4.7 and 119 ppm, respectively. The chemical shifts for the proton dimension were referenced relative to 4,4-dimethyl-4-silapentane-1-sulfonic acid (DSS), and the nitrogen and carbon dimensions were indirectly referenced using the respective gyro-magnetic ratios relative to that of a proton (*Wishart et al., 1995*).

Backbone chemical shifts were assigned using HNCO, HNCA, HNHA, CBCA(CO)NH, HNCACB and $^{15}N$ HSQC-NOESY-$^{15}N$ HSQC experiments (*Kay et al., 2011*). Side chain assignments were completed with $^1H$-$^{13}C$ HSQC, $^{13}C$ (CT)-HSQC, H(CC)(CO)NH and CC(CO)NH experiments (*Grzesiek et al., 1993*). The (CT)-HSQC spectrum was recorded with 28 ms of constant time $^{13}C$ evolution. TOCSY mixing time for the H(CC)(CO)NH and CC(CO)NH experiments was set at 16 ms. $^1H$-$^1H$ distance restraints were measured from $^{15}N$ edited NOESY and $^{13}C$ edited NOESY spectra. $^{15}N$ edited NOESY data were acquired with 80 ms mixing times, whereas $^{13}C$ edited NOESY spectra were acquired with 120 ms mixing times. The weakly aligned sample for the RDC measurements was prepared by mixing the protein sample with a final concentration of 17 mg/ml of Pf1 phage (Asla Biotech, Riga, Latvia) (*Hansen et al., 1998*). The backbone $^1D_{NH}$ and $^1D_{C\alpha H\alpha}$ residual dipolar couplings were measured from 2D IPAP-HSQC (*Ottiger et al., 1998*) and 3D IPAP-J-HNCO(CA) (*Yang et al., 1998*) experiments, respectively.

The backbone dihedral angle (φ and ψ) restraints and the side chain $\chi^1$ angles were estimated from chemical shifts using TALOS+ (*Shen et al., 2009*) and PREDITOR (*Berjanskii et al., 2006*), respectively. In total, 15 $\chi^1$ angles were selected based on a high confidence limit (>85%). The proton–proton

distance restraints were obtained from the peak intensities of the NOESY experiments, where NOE intensities are defined as strong (1.8–2.5 Å), medium (1.8–3.0 Å), and weak (1.8–5.0 Å). Hydrogen bond restraints were also implemented for residues in the helical segment, as judged by the presence of sequential $H_N$-$H_N$ NOEs and high helical propensity by TALOS+. Two distance restraints were defined for each hydrogen bond: 1.8–2.0 Å for the HN-O distance and 2.7–3.0 Å for the N-O distance (*Guntert et al., 1998*). The ten lowest energy structures satisfying the NOE, dihedral angle and H-bond restraints were obtained from an extended chain using standard high-temperature annealing implemented in CNS (v1.3) (*Brunger, 2007*). Next, each of these structures were refined with NH RDCs and then with Cα-Hα RDCs. During this refinement step, the temperature was decreased from 1000 to 0 K, and the NH RDC restraint potential was ramped from 0.1 to 5 kcal mol$^{-1}$ Hz$^{-2}$. Initial estimates of the axial ($D_a$) and rhombicity ($R_h$) of the alignment tensor were obtained using the program PALES (*Zweckstetter and Bax, 2000*). The force constant for the NOE and dihedral retraints were fixed at 50 kcal mol$^{-1}$ Å$^{-2}$ and 200 kcal mol$^{-1}$ rad$^{-2}$, respectively. 50 structures from each of the first 10 models were calculated. The lowest energy structure obtained from each run was subsequently refined using Cα-Hα RDCs in addition to the other restraints. The Cα-Hα RDCs were normalized to the NH RDC values (*Clore et al., 1998*). Sidechains coordinating the Ca$^{2+}$ ion in EF hand 1 (residues 23, 25, 27 and 34) and EF hand 2 (*O'Shea et al., 1991*; *Zhang et al., 1995*; *Ahmadian et al., 2002*) were implemented as distance restraints. A distance of 2.2–3 Å was defined for each of the O–Ca$^{2+}$ restraints. During this step, the temperature was decreased from 500 to 0 K, and the RDC restraint potential was ramped from 0.001 to 0.2 kcal mol$^{-1}$ Hz$^{-2}$. Ten representative structures were selected, and the quality of the selected structures was assessed using PROCHECK-NMR (*Laskowski et al., 1996*) and Protein Structure Validation Software suit v1.4 (*Bhattacharya et al., 2007*). A summary of the structural statistics is presented in *Table 4*.

## Acknowledgements

We would like to thank members of the Roose and Kuriyan labs including Julie Zorn and Aaron Cantor for critical comments on the manuscript, and scientists at Beamline 8.2.2 at Lawrence Berkeley National Laboratory for crystallography expertise.

## Additional information

### Competing interests

JK: Senior editor, *eLife*. The other authors declare that no competing interests exist.

### Funding

| Funder | Grant reference number | Author |
|---|---|---|
| National Institutes of Health | P01 AI091580 | Jeroen P Roose, John Kuriyan |
| Howard Hughes Medical Institute | | John Kuriyan, Tiago Barros |
| National Cancer Institute | U54CA143874 | Jeroen P Roose |
| National Institutes of Health | 5F32GM095149-03 | Jeffrey S Iwig |
| National Institutes of Health | R56-AI095292 | Jeroen P Roose |
| National Institutes of Health | R03AR062783 | Andre Limnander |
| Gabrielle's Angel Foundation | | Jeroen P Roose |

The funders had no role in study design, data collection and interpretation, or the decision to submit the work for publication.

### Author contributions

JSI, YV, RD, AL, TB, Conception and design, Acquisition of data, Analysis and interpretation of data, Drafting or revising the article; YC, Acquisition of data, Analysis and interpretation of data, Drafting or revising the article; JGP, Conception and design, Acquisition of data, Contributed unpublished essential data or reagents; DEW, JPR, JK, Conception and design, Analysis and interpretation of data, Drafting or revising the article

## Additional files

### Major dataset

The following datasets were generated:

| Author(s) | Year | Dataset title | Dataset ID and/or URL | Database, license, and accessibility information |
|---|---|---|---|---|
| Iwig J, Vercoulen Y, Das R, Barros T, Limnander A, Che Y, et al. | 2013 | Autoinhibited state of the Ras-specific exchange factor RasGRP1 | 4L9M; http://www.rcsb.org/pdb/search/structidSearch.do?structureId=4L9M | Publicly available at the Protein Data Bank (http://www.rcsb.org/pdb/). |
| Iwig J, Vercoulen Y, Das R, Barros T, Limnander A, Che Y, et al. | 2013 | Structure of C-terminal coiled coil of RasGRP1 | 4L9U; http://www.rcsb.org/pdb/search/structidSearch.do?structureId=4L9U | Publicly available at the Protein Data Bank (http://www.rcsb.org/pdb/). |
| Iwig J, Vercoulen Y, Das R, Barros T, Limnander A, Che Y, et al. | 2013 | Solution structure of the RasGRP2 EF hands bound to calcium | 2MA2; http://www.rcsb.org/pdb/search/structidSearch.do?structureId=2MA2 | Publicly available at the Protein Data Bank (http://www.rcsb.org/pdb/). |

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
