## [Decision Letter]

Thank you for sending your work entitled “Structural analysis of autoinhibition in the Ras-specific exchange factor RasGRP1” for consideration at *eLife*. Your article has been favorably evaluated by a Senior editor, Detlef Weigel, and 2 reviewers, one of whom is a member of our Board of Reviewing Editors.

The Reviewing editor and the other reviewer discussed their comments before we reached this decision, and the Reviewing editor has assembled the following comments to help you prepare a revised submission.

Both reviewers believe that your data provide significant and new insights into Ras biology and should be published in *eLife*. Before the paper can be published, however, several concerns should be addressed in a revised manuscript:

1) You state that the helical hairpin and the Cdc25 domain in toto is poised to bind Ras. Why, then, is the activity of RasGRP so extremely slow, much slower than any of the other Cdc25 domain proteins? Is this a matter of a low affinity? The authors cite the open conformation of Epac or Cdc25, but these have a high activity – that’s why comparison with those does not make sense. Please explain.

Why do the authors emphatically exclude an allosteric mechanism? Such a mechanism could easily be tested biochemically, as such assays have extensively been performed in the Kuriyan lab. Please explain why an allosteric mechanism can be excluded.

2) Is the CEC construct dimeric as compared to CAT being monomeric? If the interface is so tight and extensive (2500 A2) as described, one would expect it to be dimeric in solution. Is it only dimeric together with the CC domain? Please explain.

3) In the cell culture experiments, where increasing amounts of RASGRP1 were used, is the resulting ERK activation obtained after stimulation with something?

4) There is a striking discrepancy in Figure 7, where increasing concentrations of RasGRP-CAT versus SOS-CAT reach similar activities in cells but are very much different in in vitro assays, without explanation. Doesn’t that mean that there is another activation mechanism in cells? Please explain.

5) The contributions of either the linker reorientation or Ca binding might be over-interpreted, as they produce rather minor effects. The real significant effect seems to be due to DAG (Figure 8). Please explain.

6) The apo form of RasPRG1 EF seems to be in a conformationally heterogeneous state. The NMR spectrum of RasGRP2 EF however indicates that this domain might be largely unfolded in the apo state. Please indicate this difference between both proteins in their apo states in the text or provide a CD spectrum of RasGRP2 EF in its apo and Ca states (similar to RasGRP1 EF) that clarifies its conformational state.

7) In Figure 3 please provide an additional overlay with the active helical hairpin from either SOS or another protein to get a feel for what it should look like.

8) Please include in the Discussion a section about the general activation mechanism, how many different pathways are involved in obtaining full activation, and if the integration of different pathways might make the activation “switch-like”.

---

## [Author Response]

*1) You state that the helical hairpin and the Cdc25 domain in toto is poised to bind Ras. Why, then, is the activity of RasGRP so extremely slow, much slower than any of the other Cdc25 domain proteins? Is this a matter of a low affinity? The authors cite the open conformation of Epac or Cdc25, but these have a high activity – that’s why comparison with those does not make sense. Please explain*.

We understand the reviewers’ confusion about the low RasGRP1 activity given the open conformation of the helical hairpin. The RasGRP1 catalytic activity in solution is likely low due the weak affinity for Ras, relative to other Cdc25 domains. Comparison of the sequences of the Cdc25 domains of SOS and RasGRP1 indicates that the SOS residues that contact the switch 1 region of Ras are not conserved in RasGRP1. We cannot rule out that that stronger activity would be observed for RasGRP1 using KRas or NRas. It should also be pointed out that activity of RasGRP^CEC^ with Ras-coupled vesicles is similar to that of SOS^cat^. We have elaborated on the differences between RasGRP1 and SOS in the text so that readers better understand why RasGRP1^cat^ activity is low.

*Why do the authors emphatically exclude an allosteric mechanism? Such a mechanism could easily be tested biochemically, as such assays have extensively been performed in the Kuriyan lab. Please explain why an allosteric mechanism can be excluded*.

As the reviewers state, our group has in the past used biochemical assays to explore the allosteric regulation of SOS. These experiments show that SOS^cat^ activity increases several hundred-fold when Ras-coupled vesicles are used as a substrate versus Ras in solution. This is due to Ras binding to the allosteric site and increasing the local concentration of SOS at the membrane, where it is more likely to encounter Ras at the active site. We do not observe a similar enhancement in nucleotide exchange activity for RasGRP1, as shown in Figure 7, and therefore conclude that RasGRP1 is unlikely to posses an equivalent Ras-binding site. SOS^cat^ activity has also been shown to depend on the nucleotide-bound state of Ras due to differences in the affinity for the allosteric site. In data that is not shown, we see no difference in nucleotide exchange activity for RasGRP1 and Ras, with GTP or GDP in solution. However, we cannot rule out that GTPases other than Ras could bind to an allosteric site on RasGRP1. We have made clarifications in the text that more explicitly state the evidence that argues against an allosteric site in RasGRP1 and moved the location of this text so that it is clear why the data is inconsistent with an allosteric site. We have also softened the language of this section to reflect the possibility of other interpretations.

*2) Is the CEC construct dimeric as compared to CAT being monomeric? If the interface is so tight and extensive (2500 A2) as described, one would expect it to be dimeric in solution. Is it only dimeric together with the CC domain? Please explain*.

The reviewers are correct that RasGRP1^CEC^ can oligomerize in solution. We actually observe multiple oligomeric states, including monomers, dimers and tetramers, for RasGRP1^CEC^, as measured by gel filtration. This distribution is concentration dependent but due to the complexity of the system it is difficult to thoroughly analyze the data. We believe the coiled coil will enforce the dimeric arrangement observed in the crystal structure. We have included gel filtration traces for RasGRP1^CEC^ in Figure 6—figure supplement 1 as well as corresponding text.

*3) In the cell culture experiments, where increasing amounts of RASGRP1 were used, is the resulting ERK activation obtained after stimulation with something*?

The cellular experiments with RasGRP1 are in unstimulated cells, except where explicitly stated. We have modified the text in the Results section to reflect this.

*4) There is a striking discrepancy in Figure 7, where increasing concentrations of RasGRP-CAT versus SOS-CAT reach similar activities in cells but are very much different in in vitro assays, without explanation. Doesn’t that mean that there is another activation mechanism in cells? Please explain*.

The in vitro and cellular assays with RasGRP1^cat^ and SOS^cat^ are actually in clear agreement. The reviewers note that in cells the two proteins reach similar levels of activation. However, perhaps it was overlooked that Figure 7 clearly shows that at low and medium expression levels SOS^cat^ is significantly more active than RasGRP1^cat^. Only at very high expression levels when the system is saturated do the activities become similar. We have previously reported (78) that strong Ras signaling induced at high SOS^cat^ expression levels results in a flattening of the dose-response curve and limits maximal levels of ERK phosphorylation due to negative feedback mechanisms in the cell. This highlights why we have chosen to use FACS for monitoring cellular activity of the different nucleotide exchange factor constructs. The differences in activity observed in vitro with Ras-coupled vesicles also show that SOS^cat^ activity is stronger than RasGRP1^cat^. At higher RasGRP1^cat^ concentrations, the nucleotide exchange activity would eventually approach that of SOS^cat^.

*5) The contributions of either the linker reorientation or Ca binding might be over-interpreted, as they produce rather minor effects. The real significant effect seems to be due to DAG (Figure 8). Please explain*.

Many of the RasGRP1 mutants analyzed in this work have statistically significant differences in their activities than the wild type protein and we believe that although these effects are not as strong as those of diacylglycerol the data do point to the importance of individual inhibitory elements in net regulation. To increase our confidence in this assertion, we have relied on both in vitro and cellular assays. Also, note that the nucleotide exchange experiment in Figure 8 that shows the large effect of diacylglycerol on activity is done under conditions with high Ras densities and saturating diacylglycerol levels, meaning that this is near the maximum effect that can be observed in this artificial system. These conditions are unlikely to be recapitulated in a cell. This is noted in the text.

*6) The apo form of RasPRG1 EF seems to be in a conformationally heterogeneous state. The NMR spectrum of RasGRP2 EF however indicates that this domain might be largely unfolded in the apo state. Please indicate this difference between both proteins in their apo states in the text or provide a CD spectrum of RasGRP2 EF in its apo and Ca states (similar to RasGRP1 EF) that clarifies its conformational state*.

The requested CD spectra for RasGRP2^EF^ in the apo and Ca-bound states have been obtained under the same conditions used for RasGRP1^EF^. Like RasGRP1^EF^, RasGRP2^EF^ shows significant helical content in the apo state and this increases upon the addition of calcium. The new spectra have been added to Figure 10—figure supplement 4 and this is now indicated in the text.

*7) In Figure 3 please provide an additional overlay with the active helical hairpin from either SOS or another protein to get a feel for what it should look like*.

We agree that this addition is important and we have added the helical hairpin from SOS in the active state (yellow) to Figure 3.

*8) Please include in the Discussion a section about the general activation mechanism, how many different pathways are involved in obtaining full activation, and if the integration of different pathways might make the activation “switch-like”*.

RasGRP1 has been observed to signal in a graded manner in T cells (Roose et al 2007). Therefore, it appears that RasGRP1 activity is not switch-like, but is tightly coupled to the concentrations of the calcium, diacylglycerol, and phosphorylation signals. Further experiments will be necessary to understand how RasGRP1 integrates all of signaling inputs, leading to the graded signaling response. We have made added these important ideas to the “Concluding Remarks” section of the text.